# Defined roles for the *Staphylococcus aureus* POT transporter DtpT in di/tripeptide uptake and glutathione utilisation inside human macrophages

Imran Khan[1], Sandy J. MacDonald[2], Sigurbjörn Markússon[3], Paige J. Kies[4], Cristina Kraemer-Zimpel[4], Callum Robson[1], Joanne L. Parker[3,5], Simon Newstead[3,5], Dave Boucher[1], Neal D. Hammer[4], Marjan Van Der Woude[6], Gavin H. Thomas[1]*

1 Department of Biology, University of York, York, United Kingdom, 2 Bioscience Technology Facility, University of York, York, United Kingdom, 3 Department of Biochemistry, University of Oxford, Oxford, United Kingdom, 4 Department of Microbiology, Genetics & Immunology, Michigan State University, East Lansing, Michigan, United States of America, 5 Kavli Institute of Nanoscience Discovery; University of Oxford, Oxford, United Kingdom, 6 Hull York Medical School, University of York, York, United Kingdom

* gavin.thomas@york.ac.uk

## Abstract

Peptides available in biological niches inhabited by the human pathogen *Staphylococcus aureus* serve as a rich source of amino acids required for growth and successful host colonisation. Uptake of peptides by *S. aureus* involves at least two transport systems: the di/tripeptide permease DtpT and the oligopeptide ABC transporter Opp3. Here we study the individual and combined functions of DtpT and Opp3 in enabling utilisation of diverse di-/tripeptides via a high-throughput phenotypic screen. We reveal that DtpT is the primary route of uptake for dipeptides, and although many peptides can be utilised via either of the two transport systems, we demonstrate a clear preference for Asp/Glu-containing peptides among DtpT substrates. To better understand the substrate preferences of DtpT, the protein was purified and reconstituted into proteoliposomes. Active transport of diverse di- and tripeptides was demonstrated, supporting the conclusions of the phenotypic screen. During this *in vitro* analysis, we discovered that DtpT could transport the biologically prevalent tripeptide glutathione (GSH). Bacterial growth assays demonstrate that *dtpT* is essential for GSH utilisation in the absence of the known glutathione transporter, Gis, identifying DtpT as the second GSH uptake system of *S. aureus*. We demonstrate that GSH transport is required by *S. aureus* for complete fitness during *in vitro* macrophage infection experiments. Finally, based on analysis of the DtpT structure and identification of key residues needed for GSH binding and transport, we suggest that GSH transport may be conserved in the DtpT orthologue of *Listeria monocytogenes*. Together, these data reveal important new functions for DtpT in the utilisation of diverse peptides and point toward a novel role for DtpT (and, potentially, other bacterial POT proteins) in glutathione acquisition during intracellular infection.

**Data availability statement:** All relevant data are in the manuscript and its supporting information files.

**Funding:** o This work was funded by grants to GHT from the Biotechnology and Biological Sciences Research Council (BBSRC) (BB/X003035/1) and similarly to DB (BB/Y009703/1) and work undertaken in the NDH lab was supported by the National Institute of Allergy and Infectious Diseases of the National Institutes of Health, R01 AI139074. IK was supported in his PhD work by a gift from the Burgess family to the University of York. The funder had no role in the study design, data collection and analysis, decision to publish or preparation of the manuscript. Please can this additional funder be added to the submission form.

**Competing interests:** The authors have declared that no competing interests exist.

## Author summary

The environments where bacterial pathogens thrive are often rich in proteins and their degradation products, including oligopeptides, which can be taken up by the bacterium and used as nutrients. Understanding how this occurs could help us find ways to tackle the growth of these pathogens during infection. Here we examine how the major human pathogen *Staphylococcus aureus* takes up oligopeptides. We demonstrate that a membrane transporter protein, DtpT, is the major route of dipeptide uptake and reveal over 100 new di-/tripeptide targets for this protein. Our findings highlight a defined role for DtpT in the accumulation of Aspartate- and Glutamate-containing peptides, which may serve as relevant nitrogen sources during infection. We also provide the first evidence that DtpT transports the prevalent human metabolite reduced glutathione, and demonstrate that DtpT functions alongside the previously identified Gis glutathione transport system to support intracellular survival of this pathogen inside macrophages. Overall, our findings provide a clear example of how substrate selectivity allows DtpT to fulfill specific biological roles in *S. aureus*, and this functional specialisation may be a common feature of homologous peptide transporters in other bacterial pathogens and across the tree of life.

## Introduction

*Staphylococcus aureus* is a Gram-positive human bacterial pathogen associated with a broad range of different disease manifestations including skin/soft tissue infections, systemic bacteraemia, endocarditis and other invasive disease states associated with significant mortality [1]. In order to meet their nutritional requirements during infection, bacterial pathogens such as *S. aureus* possess diverse transport systems which allow for the selective uptake of nutrients present within the host. Understanding the intrinsic factors contributing to bacterial fitness, colonisation and infection in relevant infection environments is essential for informing the design of novel therapeutic approaches. While the importance of some individual transporters in successful colonisation is well documented – particularly those implicated in the acquisition of key carbon sources, amino acids and metals [2–5] – the roles of many transport systems and their contributions toward infection remain understudied.

Peptides serve as an abundant source of amino acids, and hence can provide carbon, nitrogen and sulphur. As such, most bacterial pathogens possess one or more routes of uptake for these molecules. Most bacterial peptide transporters studied to-date are members of the ATP-Binding Cassette (ABC) family of importers or secondary active transporters belonging to the Major Facilitator Superfamily (MFS), with the latter being further divided into a small number of functionally distinct groups including the proton-dependant oligopeptide transporters, or POTs [6]. ABC peptide importers are unique to prokaryotes and are capable of recognising and transporting diverse oligopeptides with high affinity, with substrates ranging from di-/tripeptides up

to 35-mers [7–10]. In addition to their canonical role in nutrient acquisition, there are examples of these transporters contributing to diverse physiological functions including cell-wall turnover [9,11] and antimicrobial peptide resistance [12,13], as well as acting as sensory units for intercellular and environmental signalling networks [14]. POTs, by comparison, are found across all domains of life and exclusively transport di- and tripeptides or chemically related molecules of a similar size [15–17]. Despite their ubiquity, the importance of bacterial POTs during infection is poorly characterised and little is known regarding their contribution to bacterial physiology outside of their general role in nutrient uptake.

To date, there are two known peptide transport systems which are widely distributed in staphylococci: the ABC importer-type system Opp3$_{ABCDF}$ (or Opp3) and the di-/tri-peptide POT transporter DtpT [18]. In the single elegant study comparing the roles of these two systems in *S. aureus*, Opp3 was demonstrated to be required for the transport of oligopeptides from 3 to 9 residues in length, while only DtpT was thought to be responsible for dipeptide uptake [19]. *S. aureus* also possesses an ABC importer system which is specific for the biologically prevalent tripeptide glutathione (Gis$_{ABCD}$, or Gis) [20]. In addition to the aforementioned systems, *S. aureus* strains can possess up to four additional Opp3-homologues, of which Opp1 and Opp2 are likely metal rather than peptide transporters [21,22]. However, Opp4 plus one additional Opp system (ACME Opp) encoded as part of the widely disseminated arginine catabolic mobile element [18] are mostly uncharacterised and could function as additional peptide transporters.

The importance of peptide utilisation in *S. aureus* infection biology is a subject of ongoing investigation. *S. aureus* secretes an arsenal of extracellular proteases with diverse functions in host immune modulation and virulence, and these enzymes contribute to the generation of peptides within the local environment which are likely to serve as a nutrient source for the bacterium [23,24]. Consistent with this, there is evidence that *S. aureus* senses protein-rich environments via Opp3 and responds to this stimulus via upregulation of extracellular proteases [25,26]. Recent work has demonstrated how the *S. aureus* protease aureolysin is capable of contributing to the degradation of collagen and highlights how growth on collagen-degradation products is decreased in an Opp3-deficient mutant [26]. These observations point toward a model in which host- and bacterial-derived peptides support the growth of *S. aureus* in an Opp3-dependent manner. Conversely, loss of DtpT – but seemingly not Opp3 – leads to attenuation in several models of invasive *S. aureus* infection including intravenous infection in mice and an endocarditis model in rabbits [23]. In spite of this, the functional characterisation of DtpT to date is limited, and the molecular basis of its role in bacterial survival during infection is unknown.

Recent work has aimed to understand the molecular basis of substrate promiscuity observed in POTs and delineate the mechanistic basis of proton-coupled peptide transport in these proteins [16,27,28]. However, little is known regarding the substrate preferences of these transporters and how this relates to cellular lifestyle/environment. Here, we aim to address this gap in our knowledge. We apply Phenotype Microarrays (BIOLOG) to compare the utilisation of varied di- and tripeptides by wild-type *S. aureus* JE2 against strains with disruptions in the known peptide transporters. This approach provides novel insights into the diversity of peptides utilised via DtpT and points toward distinct amino acid preferences among DtpT substrates. Further cell-based and biochemical assays serve to both support these findings and further interrogate the specificity of the system. Finally, we extend these assays to demonstrate how DtpT also serves as a transporter for the biologically prevalent metabolite reduced glutathione (GSH) and demonstrate how glutathione transport mediated by DtpT and Gis is a determinant of bacterial survival inside human macrophages.

## Results

### Phenotypic screening defines the substrate landscapes of DtpT and Opp3A

In order to investigate the roles of the two known peptide transporters in *S. aureus*, we generated an isogenic series of strains in the USA300-derivative strain JE2 with disruptions of *dtpT* and *opp3A,* as well as a double-mutant strain. With these strains in hand, we exploited the power of phenotype microarrays (PMs) as a simple high-throughput approach for comparing the abilities of bacterial strains to catabolise diverse substrates, including di- and tripeptides (Biolog PM Plates 6–8) [29,30]. Collectively, these plates allowed us to compare utilisation for 282 different peptides, comprising: 246

standard dipeptides (of a possible 400), 1 hydroxyproline-containing dipeptide, 7 β-bonded dipeptides, 2 γ-bonded dipeptides, 12 D-amino acid-containing dipeptides and 14 tripeptides. In this assay, a given peptide is supplied as the primary nitrogen source, and cellular metabolism is measured via reduction of a redox-sensitive colourimetric dye, quantified over time to produce a signal curve for each strain/peptide combination. A positive metabolic signal is indicative of peptide utilisation enabled by a valid route of uptake and downstream catabolic pathways, while intermediate/weak signals are likely to result from poor uptake (or a complete lack thereof) or incomplete catabolism following internalisation.

Based on the signal curves produced from our PM analysis, we could observe a series of distinct utilisation patterns across our four test strains (S1-S4 Data files). To identify commonalities in the data, we subjected the raw signal curves for each peptide to unbiased cluster analysis and dimensional reduction using Uniform Manifold Approximation and Projection (UMAP) (Fig 1). These analytical methods provided complementary outputs, with peptides belonging to a common cluster demonstrating close spatial organisation in the final UMAP plot (Fig 1A-1C).

Visual confirmation of the clusters suggested that each encompasses a distinct pattern which reflects differences in the routes of peptide utilisation (Fig 1B and 1D). Cluster 1 (n = 15) contains peptides for which a strong metabolic signal was observed for JE2 and all three mutant strains, suggesting – rather intriguingly – that neither of the two known peptide transporters of *S. aureus* is required for their uptake. In contrast, utilisation of peptides in cluster 2 (n = 41) was maintained

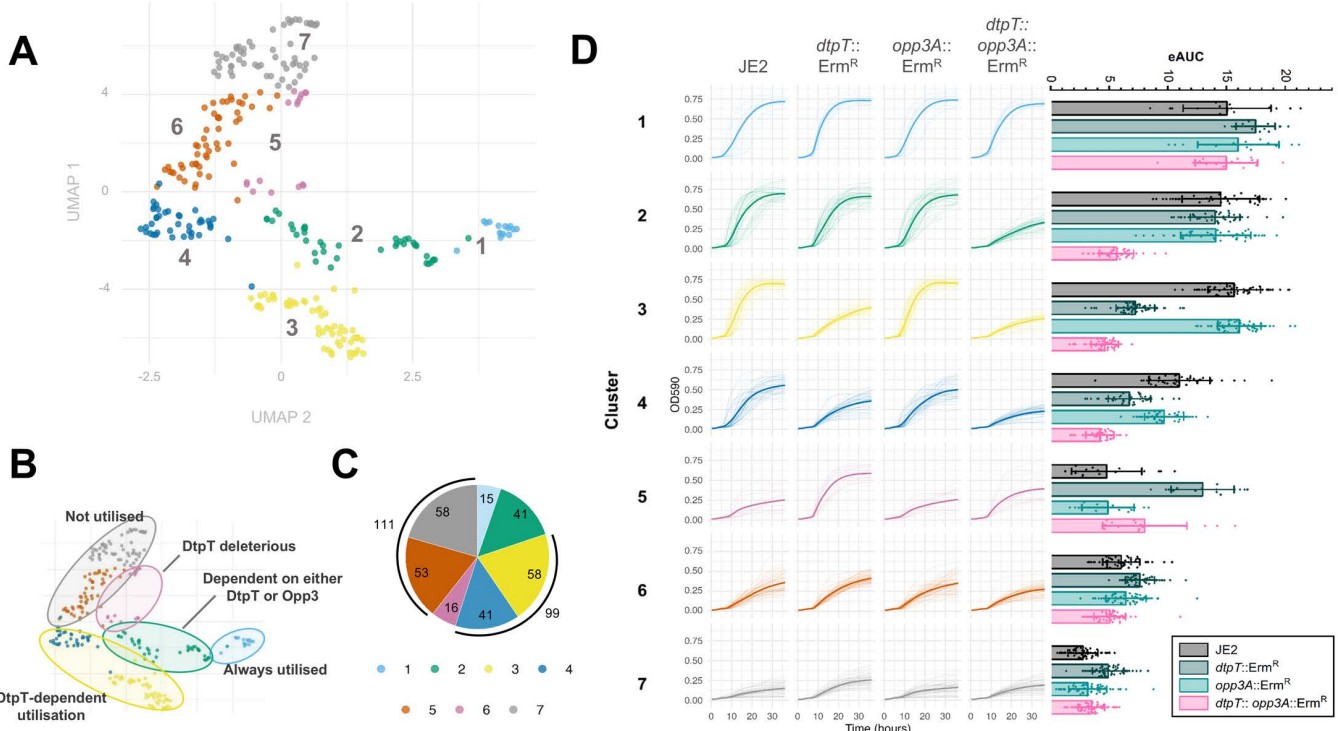

**Fig 1. UMAP and cluster analysis reveal differential peptide utilisation patterns across *dtpT* and *opp3A* mutant strains.** (A) PM signal curves (OD590 (representing reduction of the redox-sensitive tetrazolium dye) corrected for OD750 (bacterial turbidity) over time) were subjected to UMAP and k-means cluster analysis (k = 7). Peptides belonging to the same cluster are grouped by colour on the UMAP plot, as labelled. (B) Regions of the UMAP plot corresponding to distinct phenotypic patterns (determined by clustering) are highlighted. (C) Pie-chart representation of the number of peptides in each of the seven clusters identified here. In each case, colours correspond to the seven clusters identified in this analysis, as depicted by the key. (D) (**Left**) Cluster analysis reveals seven strain-specific peptide utilisation patterns. All signal curves for each cluster are overlaid and separated by strain. Mean signal curves for each strain are indicated by the bold line. (**Right**) Empirical area under the curve (eAUC) was calculated for all signal curves. Bars show the mean eAUC for each strain within a given cluster ± standard deviation in each case.

in single mutant strains lacking either *dtpT* or *opp3A* but abolished in the double-mutant strain, suggesting that these peptides can be transported by either of our two systems. Clusters 3 and 4 (n = 99) both present patterns wherein utilisation is abolished for the *dtpT* mutant and the double mutant, but not the *opp3* mutant, suggesting DtpT serves as the primary route of uptake for these peptides. Collectively, these two clusters account for more than one-third of the tested peptides, with those in cluster 3 (n = 58) demonstrating a more pronounced phenotype (that is, a greater difference in signal between "positive" and "negative" utilisation) when compared to those in cluster 4 (n = 41). Such a distinction likely results from either a higher rate of internalisation by DtpT or more productive downstream catabolism following uptake for cluster 3 peptides.

The three remaining clusters contain peptides for which a weak metabolic signal was observed in the wild-type JE2 strain. Cluster 5 (n = 16) encompasses peptides for which, quite surprisingly, metabolism is permitted only in mutant strains lacking *dtpT*. While the exact mechanistic basis for this pattern is unclear, we postulate that the DtpT-mediated internalisation of these peptides may be cytotoxic under the conditions tested. However, a strong signal in the *dtpT* mutant strain suggests an alternate route of internalisation for which toxicity is not observed, perhaps due to a lower overall flux which prevents toxic over-accumulation of the peptide. The remaining peptides each gave a weak/intermediate signal or were not utilised at all or across the four strains tested here, as observed in cluster 6 (n = 58) and cluster 7 (n = 53) respectively. These phenotypic patterns suggest that the supplied peptide alone cannot be effectively utilised for cellular metabolism. We do note that the mean eAUC values recorded for cluster 6 are slightly higher than those in cluster 7 across the four strains, perhaps indicating that some peptides in cluster 6 can be utilised to produce an intermediate metabolic signal.

Notably, our analysis did not identify a cluster (nor even any individual peptides) for which the patterns of metabolism suggested that utilisation was solely dependent on Opp3, which is perhaps expected given that this system is primarily implicated in the utilisation of larger oligopeptides (n ≥ 3) [19,26]. Taken together, our findings implicate DtpT in the utilisation of at least 140 of the peptides assessed here, with 41 of these being recognised by both peptide transport systems.

## Cluster analysis demonstrates amino acid preference among putative DtpT substrates

Next, we aimed to determine whether specific amino acids were enriched within the peptides of each cluster. Given the limited panel of tripeptides assessed, we focussed our analysis on the 246 standard dipeptides present in the assay. First, we plotted a grid to identify potential cluster-specific trends in regard to the N- and C-terminal amino acid composition of dipeptides included in the phenotype microarray analysis (Fig 2). Then, we examined any statistically meaningful relationships by applying a Fisher exact test (Fig 3A-3D). The strongest association was identified between glutamine-containing peptides and cluster 1, with the vast majority (9/13) of Gln-containing peptides assessed here belonging to this cluster (Fig 3A). Such an observation strongly suggests that an alternate route of uptake exists for Gln-containing peptides in *S. aureus,* which is independent of both DtpT and Opp3. Another clear relationship was observed between tyrosine-containing dipeptides and cluster 7 (Fig 3C), suggesting that tyrosine-containing peptides are either poorly transported or metabolised by *S. aureus* under the conditions tested here. Tantalisingly, we observe that aspartate and glutamate-containing peptides are both strongly enriched within the DtpT-specific cluster 3 (Fig 3D). This preference is seemingly independent of amino acid arrangement, with comparable frequencies for DtpT-specific utilisation in the N- or C-terminal positions (Fig 2). Our findings provide compelling evidence that Asp- and Glu-containing peptides are preferred transport targets for DtpT, in-turn suggesting that utilisation of these peptides is a distinct function of the transporter.

In addition, we found that arginine-containing peptides were highly enriched in cluster 2 (Fig 3B), implicating both peptide transport systems in the utilisation of these peptides. Arginine auxotrophy in *S. aureus* USA300 was previously demonstrated using chemically defined medium containing glucose (CDMG) and the same medium lacking arginine (CDMG-R) [31,32]. Here, we exploit this auxotrophy to verify the DtpT/Opp3A-dependence patterns of three arginine-containing peptides as observed in our phenotypic screen. Specifically, we supplemented CDMG-R with two cluster 3 peptides (Arg-Asp and Arg-Phe) and one cluster 2 peptide (Arg-Ala) and separately inoculated these media with JE2

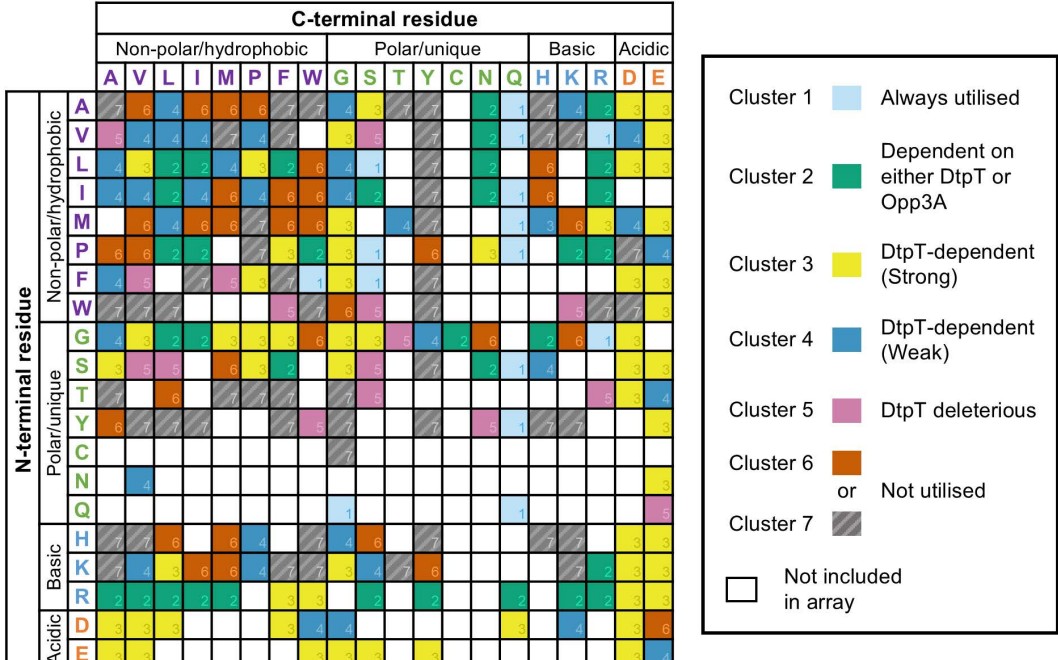

**Fig 2. Phenotype microarrays reveal differential routes of uptake for diverse dipeptides in *S. aureus*.** Grid representation of all possible dipeptide structures arranged with N-terminal amino acids on the Y axis and C-terminal amino acids on the X axis. Amino acids are displayed as single-letter symbols and grouped according to their chemical properties. Coloured cells correspond to peptides included in the PM analysis and their corresponding utilisation patterns, as indicated by the key.

and the three mutant-derivative strains. As expected, robust growth for all strains was observed in CDMG, while none of the strains were able to grow in CDMG-R (Figs 3E and S1A). Supplementing with Arg-Asp (cluster 3) restored growth in a DtpT-dependent manner, while Arg-Ala (cluster 2) restored growth in all but the double mutant strain (Figs 3E and S1A). These patterns support the findings of our phenotypic screen. Unexpectedly, supplementing with Arg-Phe (cluster 3) restored growth in all but the double mutant strain similarly to Arg-Ala (cluster 2). Such deviation is likely indicative of differences in the sensitivity of the two approaches utilised here; that is, Opp3-mediated transport of Arg-Phe is sufficient to relieve Arg auxotrophy when supplied at 1 mM, but too limited to support metabolism as a sole nitrogen source under PM conditions. Finally, to further verify the role of *dtpT* in the utilisation of these three peptides, we cloned the gene into a low-copy number vector and were able to observe genetic complementation of growth in each case (S1B Fig). Hence, we confirm that DtpT serves as a route of entry for cluster 2 and cluster 3 peptides, supporting the findings of our PM assays.

## DtpT transports peptides with diverse chemical properties

Peptide utilisation (as quantified by phenotype microarrays) produces a composite phenotype representing the combined consequences of peptide internalisation and catabolism, both of which are required to drive cellular metabolism and produce a positive signal. As a result, peptides with limited uptake may produce a positive PM signal if efficiently catabolised while other peptides may be rapidly internalised (by DtpT or otherwise) without producing a positive PM signal due to incomplete catabolic pathways.

To compare DtpT-mediated transport of peptides directly, we expressed and purified *S. aureus* DtpT recombinantly from *Escherichia coli* cells (S2 Fig) and reconstituted the pure protein into liposomes (S3A-S3B Fig) using methods previously applied to study the *Staphylococcus hominis* DtpT orthologue PepT$_{Sh}$ [33]. Transport of di-/tripeptide substrates

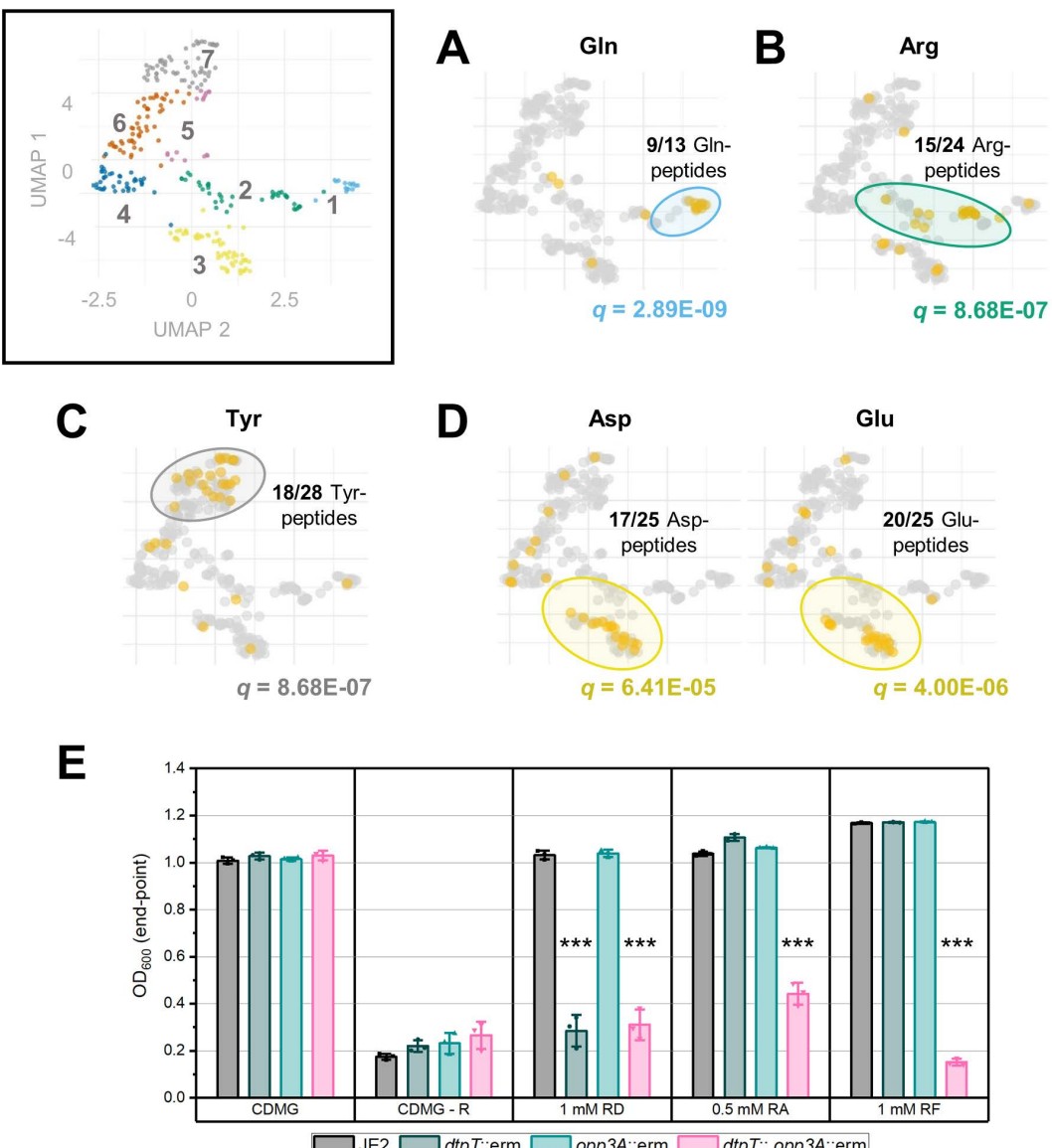

**Fig 3. Specific amino acids are enriched in peptides belonging to a common utilisation pattern.** (A – D) Enriched amino acids within each cluster were identified via a Fisher exact test with FDR q-value correction. Plots show the position of peptides containing strongly enriched amino acids ($q \leq$ 1E-03) overlayed on the original UMAP plot (as labelled). The value of $q$ is given in each case. (E) Growth of *S. aureus* strain JE2 and mutant derivatives in CDMG – R supplemented with three dipeptides, as labelled. Arg-Asp (RD) and Arg-Phe (RF) are each cluster 3 peptides, while Arg-Ala (RA) belongs to cluster 2. Bars show the mean end-point $OD_{600}$ value after 24 hours of growth ± standard deviation. End-point OD is reflective of the maximal cell density observed over this period. *** $p < 0.001$; unpaired t-test (vs JE2).

was then assessed by utilising the pH-sensitive dye pyranine, as described previously [28]. Briefly, peptide transport was assayed by monitoring the acidification of the liposome lumen and quantified over time for each peptide after induction of a negative-inside membrane potential (ΔΨ) (Fig 4A). To validate this methodology, we confirmed that peptide transport is dependent on the presence of both a valid substrate and ΔΨ, as evidenced by the lack of strong acidification when either of these components is absent (S3C Fig) and were then able to measure transport for a range of selected di- and tripeptides (Fig 3B-3C).

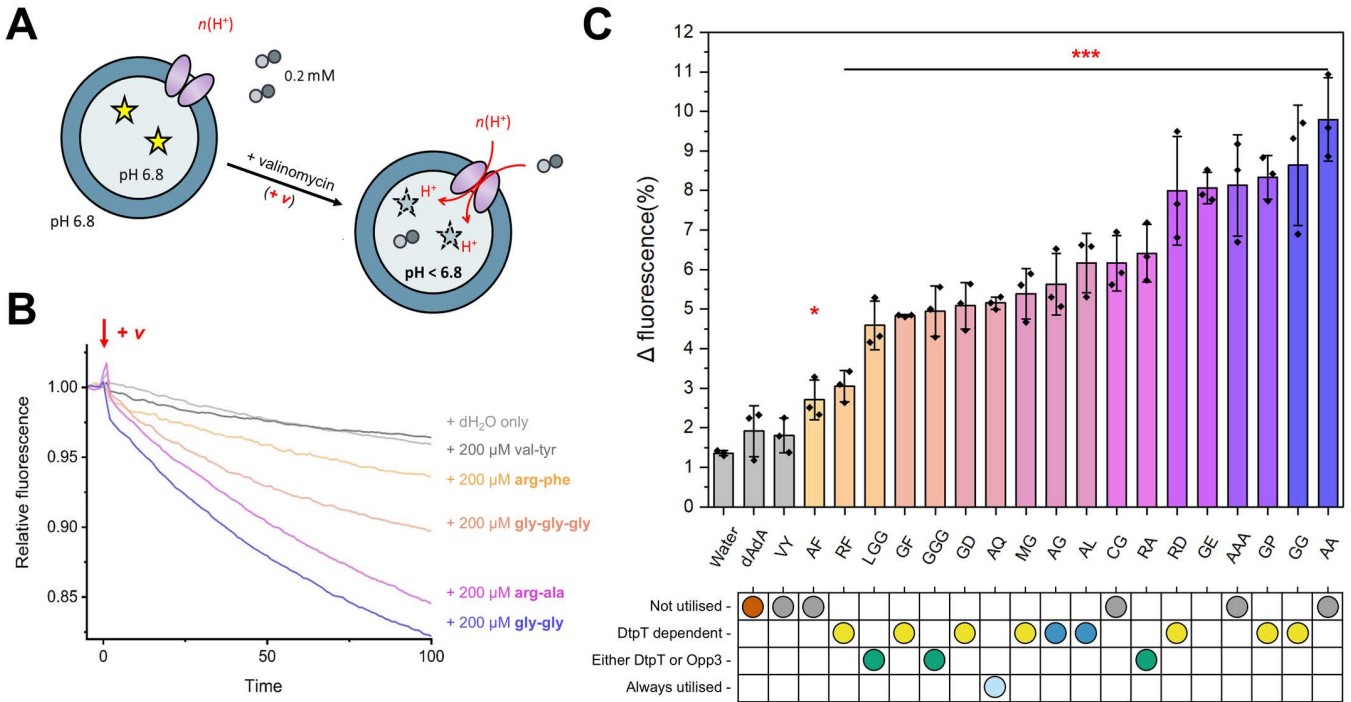

**Fig 4. DtpT transports structurally diverse di-/tripeptides.** (**A**) DtpT was reconstituted into liposomes containing pyranine (yellow stars). Upon induction of negative-inside ΔΨ by addition of valinomycin, DtpT transports peptides (small circles) from the external environment into the liposome lumen. Peptide transport is proton-coupled, leading to acidification of the lumen and a decrease in the pyranine fluorescence (ex. 460/ emm. 510). (**B**) Exemplary transport data for 5 peptides normalised immediately before the addition of valinomycin (red arrow) for ease of comparison. Curves show the mean of three replicates (error bars have been omitted for visibility). (**C**) Transport of 20 peptides was compared by quantifying the change in pyranine fluorescence over the first 30 seconds of the assay. Bars indicate the mean of three replicates ± standard deviation. Coloured bars indicate a change in fluorescence which differs significantly from the no peptide ("Water") control. Peptides are labelled according to their single-letter amino acid code (dAdA = d-ala-d-ala). *$p < 0.05$, ***$p < 0.001$; Unpaired t-test (vs Water). Corresponding phenotypic patterns identified by PM analysis for each peptide are shown below and coloured (as established in Fig 1).

By assessing transport of 20 peptides with diverse chemical properties, we were able to confirm DtpT-mediated transport for 12 of the putative substrates identified by our phenotype microarrays (Fig 4C). Notably, we confirm DtpT-mediated transport of the three Asp/Glu-containing dipeptides tested here, supporting the previous observation that peptides in this category are preferred DtpT substrates. We also confirmed the transport of Arg-Phe, Arg-Ala and Arg-Asp by DtpT, as already demonstrated in our Arg-auxotrophy experiments. Overall, there is good agreement between our liposome-based transport assays and whole-cell peptide utilisation methods (Fig 4C). We do note that some peptides are transported by DtpT which did not sustain metabolism of *S. aureus* in the phenotype microarray assay, namely; Ala-Ala, Ala-Ala-Ala and Cys-Gly. Such an observation suggests that internalisation is not the limiting factor in utilisation of these peptides.

Additionally, our transport assays shed new light on the molecular factors contributing toward substrate preference in this transporter. For instance, our data demonstrate how many favourable DtpT substrates incorporate acidic amino acids (Arg-Asp and Gly-Glu) or amino acids with small, uncharged side chains (Ala-Ala, Gly-Gly and Gly-Pro) (Fig 4C). In contrast, we demonstrate that dipeptides containing aromatic residues in position 2 generally demonstrate poorer transport when compared to those with less bulky sidechains in the equivalent position. Poor internalisation of these substrates (e.g., Val-Tyr and Ala-Phe) by DtpT is consistent with their lack of utilisation in PM assays (Fig 4C). Overall, our transport assays demonstrate that DtpT is highly promiscuous and capable of transporting substrates with a range of chemical structures and properties, which is similarly noted for other POT transporters studied previously [16,28].

## DtpT is a functional glutathione permease

Reduced glutathione (GSH) is a biologically important tripeptide metabolite with diverse roles in many organisms, primarily serving as an antioxidant in eukaryotes and some prokaryotes [34]. GSH consists of a Cys-Gly dipeptide conjugated to glutamate via a γ-linked peptide bond between the glutamate side chain and amino terminus of the cysteine (Fig 5A). Given that DtpT recognises and transports Cys-Gly (Fig 4C), we predicted that DtpT may also transport GSH and that this

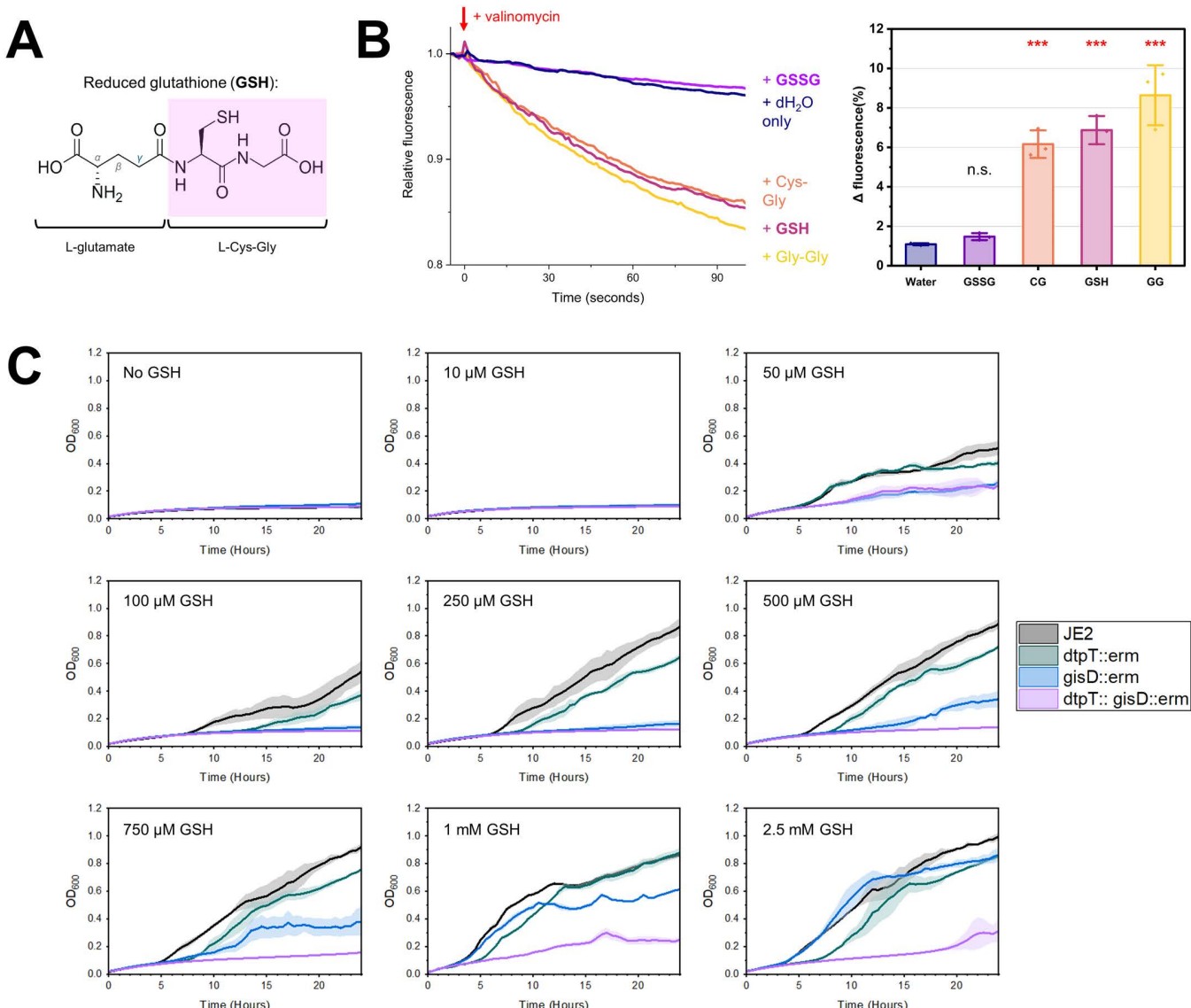

**Fig 5. DtpT transports reduced glutathione (GSH).** (**A**) Chemical structure of GSH. (**B**) (**Left**) Transport of GSH and GSSG was assessed via pyranine transport assays. Transport assay curves are normalised immediately before the addition of valinomycin (red arrow) for ease of comparison. Curves show the mean of three replicates. (**Right**) summarised transport data. Bars show the mean of three replicates ± standard deviation. ***$p < 0.001$; Unpaired t-test (vs Water). (**C**) Growth of *S. aureus* strain JE2 and mutant derivatives was assessed in CDMG – s supplemented with varying concentrations of GSH, as indicated. In each case, strains were grown over 24 hours and $OD_{600}$ was measured every 30 minutes for three biological replicates (0.05 mM and 1 mM) or four biological replicates (all other concentrations). Curves indicate the mean values and standard deviation is indicated by the shaded area in each case.

activity may be relevant in biological environments where glutathione is enriched. To test this, we measured DtpT transport activity with either GSH or oxidised glutathione (GSSG) (Fig 5B). We demonstrate that GSH is transported by DtpT, with acidification of the liposome lumen over the first 30 seconds comparable to that observed for Cys-Gly. We did not observe transport of GSSG in this assay, likely due to the larger structure of GSSG compared to GSH.

The Gis system is an ABC-type importer which has recently been characterised as a route of glutathione acquisition in *S. aureus* and is the only route of uptake for both GSH and GSSG identified in this bacterium to date [20]. However, under conditions where GSH utilisation is essential, the growth of a *gis*-deficient mutant strain is restored to near wild-type levels when GSH is supplied at concentrations greater than 0.5 mM [20]. This implies the existence of at least one additional route of GSH uptake.

To determine whether DtpT fulfils this role, isogenic mutants of *dtpT* and *gisD,* as well as a double mutant, were grown under conditions where glutathione was supplied as the sole sulphur source. At low GSH concentrations (≤250 µM), we found that the growth of *S. aureus* is limited and solely dependent on the Gis system (Fig 5C). Both single mutant strains were able to grow at concentrations ≥500 µM, while the double mutant remained non-viable under these conditions (Fig 5C). In comparison, we found that growth in the presence of GSSG was solely dependent on Gis (S4B Fig). We also found that *in trans* complementation with *dtpT* was sufficient to fully restore growth of the *dtpT* single-mutant and partially restore growth in the double mutant strain in the presence of 1 mM GSH (S4A Fig). Finally, given that GSH oxidises spontaneously over time in aerated environments, we repeated the growth experiment with 1 mM GSH under anaerobic conditions to limit changes in the oxidation state of GSH during growth (S4C Fig). Again, we observed phenotypes which were consistent with those observed during aerobic growth. Taken together, these observations suggest that Gis is a high-affinity GSH uptake system active at low micromolar concentrations, while both Gis and DtpT contribute to GSH uptake at millimolar concentrations.

### Glutathione transport contributes to bacterial survival inside macrophages

Although *S. aureus* was historically considered an extracellular pathogen, recent work has demonstrated that this bacterium survives inside phagocytic cells, highlighting how its intracellular lifestyle may contribute to dissemination and immune evasion during systemic infection [35,36]. To date, there is a poor understanding of the molecular factors governing fitness in these intracellular environments, particularly regarding nutrient availability and acquisition. Cytosolic concentrations of GSH in human cells range from 1 to 10 mM depending on the cell type, making it the most abundant intracellular non-protein thiol [37,38]. Based on this, we hypothesised that glutathione utilisation may contribute toward the fitness of *S. aureus* in intracellular environments.

Differentiated THP1 macrophage-like cells and human monocyte-derived macrophages (hMDMs) were each infected with either JE2 or mutant derivatives of *dtpT* and/or *gisD*, and bacterial survival was quantified over the first 8 hours post-infection (Fig 6). To account for any changes in fitness resulting from the inclusion of the *bursa aurealis* transposon insertion sequence, we also assessed the survival of a mutant strain carrying the transposon in a presumed phenotypically null locus corresponding to a predicted IS3-family transposase gene (SAUSA300_0060). In THP-1 cells, survival of both the *dtpT* and *gisD* single-mutant strains was reduced slightly on-average at 8 hours but did not differ significantly when compared to the wild-type. In comparison, survival of the *dtpT gisD* double mutant strain was significantly lower at this time-point, being reduced more than two-fold relative to the wild-type (Fig 6A). Similarly, survival of both the *gisD* single-mutant and *dtpT gisD* double-mutant strains was significantly reduced at this time-point in hMDMs derived from eight independent donors (Fig 6B). Overall, these data suggest that GSH transport is required for complete fitness of *S. aureus* during macrophage infection, though the presence of either *dtpT* or *gisD* is seemingly sufficient to restore survival to near wild type levels.

In eukaryotic systems, glutathione acts as an important cellular antioxidant and contributes to varied physiological processes including metal homeostasis and alleviation of lipid oxidation stress, ultimately contributing to immune cell function [39,34]. We, therefore, wondered whether bacterial sequestration of glutathione may serve as a strategy to disrupt healthy

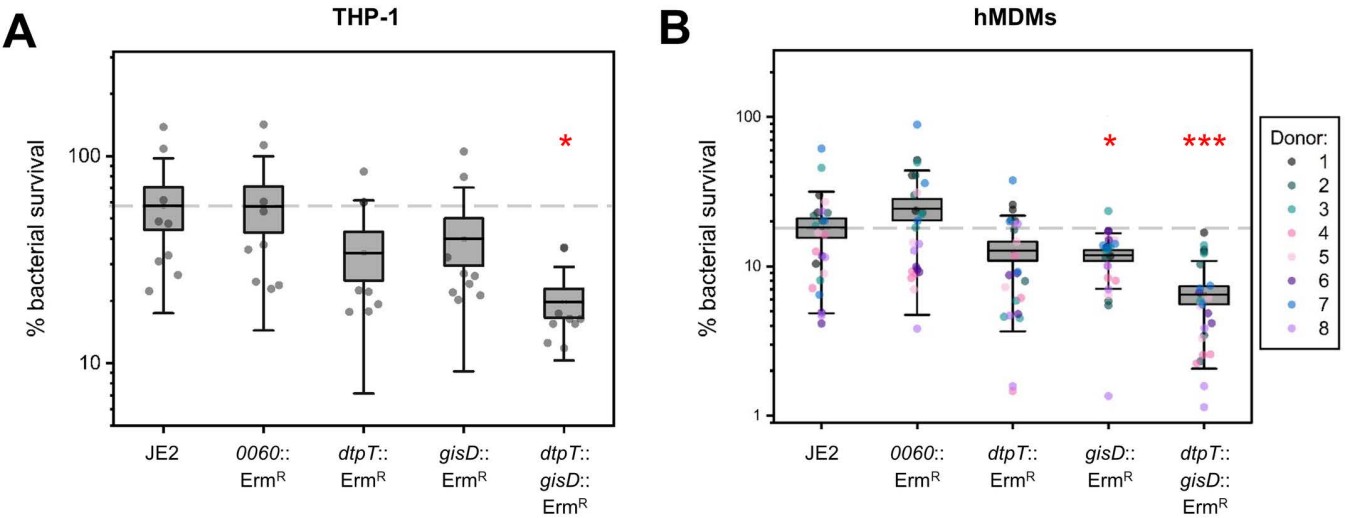

**Fig 6. A glutathione transport-null double-mutant demonstrates reduced survival inside macrophages.** (A - B) Bacterial % survival of JE2 and mutant derivatives after 8 hours post-infection in THP1 macrophage-like cells (**A**; *n* = 9) and hMDM cells (**B**; *n* = 8 donors, each infected in biological triplicate). In each case, survival is calculated as a percentage of the total bacterial uptake. Boxes indicate the mean and standard error ± standard deviation. A dashed line indicates the mean % survival of the wild-type JE2. *$p < 0.05$, **$p < 0.01$, ***$p < 0.001$; paired-sample t-test (against JE2).

phagocyte function and thereby promote bacterial survival. On average, we found no differences in cytotoxicity between the tested strains at 4 hours post-infection (S5A Fig). In contrast, we observed that THP1 cells infected with the GSH transporter-null double-mutant strain demonstrated increased cell death at 8 hours relative to the wild-type (S5B Fig). To investigate this further, we quantified the production of inflammatory cytokines at 6 hours post-infection, representing the mid-point between these two states. At this time point, cells infected with the double-mutant strain produced significantly lower levels of both IL-1β and IL6, though production of TNFα was seemingly unaffected (S5C Fig). While preliminary, these observations suggest that glutathione transporters allow *S. aureus* to modulate the macrophage pro-inflammatory status and maintain the intracellular niche by ultimately prolonging host-cell survival.

## Glutathione transport is not required during systemic murine infection

Our data indicates a role for bacterial GSH uptake in intracellular environments, but the contribution of this process toward fitness during infection is not yet understood. Previous work has demonstrated that disruption of *gis* does not significantly attenuate *S. aureus* JE2 during systemic infection in a murine model [20]. Such an observation may be explained by the capacity of DtpT to facilitate GSH utilisation at concentrations encountered *in vivo*. To determine whether *dtpT* contributes toward fitness in this model, we carried out systemic infections with wild type JE2 and an isogenic mutant of *dtpT*, as well as a *dtpT/gis* double mutant strain (see S1 File). In single-strain infection experiments, neither mutant demonstrated a pronounced difference in bacterial burden in the heart or kidneys at 96 hours post-infection. We did, however, observe significant attenuation of the *dtpT* mutant strain in the liver compared to the wild type at this time point (S6A Fig), though we failed to observe this phenotype for the *dtpT/gis* double mutant strain (S6B Fig). In a competitive infection assay, wild type JE2 did not outcompete the *dtpT* mutant strain in the liver (competitive index ≈ 1) but did outcompete a *gisB* mutant (S6C Fig). The competitive index of JE2 was greater against the *dtpT/gis* double-mutant strain on-average when compared to either of the *dtpT* or *gisB* single mutants, though this trend was not statistically significant (S6C Fig). Ultimately, our data suggest that glutathione uptake may contribute toward bacterial survival in the liver, though alternative nutrient sulphur sources available to the bacterium are seemingly able to compensate for the inability to utilise GSH in this environment.

## Recognition of GSH by DtpT in a vertical conformation during transport

Based on our discovery of physiologically relevant GSH transport through DtpT, we next sought to understand more about how this important tripeptide is recognised during transport. Previously, we demonstrated how the peptide-conjugated thioalcohol S-[1-(2-hydroxyethyl)-1-methylbutyl]-L-cysteinylglycine (S-Cys-Gly-3M3SH) is accommodated in an unconventional vertical conformation during transport by the *Staphylococcus hominis* POT PepT$_{Sh}$ [33]. Given the close sequence homology between this protein and DtpT (87% sequence identity; including all residues previously implicated in substrate binding) (S7 Fig) and the conserved L-Cys-Gly dipeptide backbone common to both substrates, we hypothesised that GSH and S-Cys-Gly-3M3SH may share a common binding mode in DtpT/PepT$_{Sh}$ during transport.

To investigate this further, we generated a structural model of DtpT in the inward-open conformation based on the existing PepT$_{Sh}$ complex structure (accession: 6EXS) and the binding mode of GSH in this protein was predicted using the protein-ligand docking tool CB-DOCK 2 (Fig 7A-7D). In the predicted conformation, the L-Cys-Gly group of GSH occupies a similar

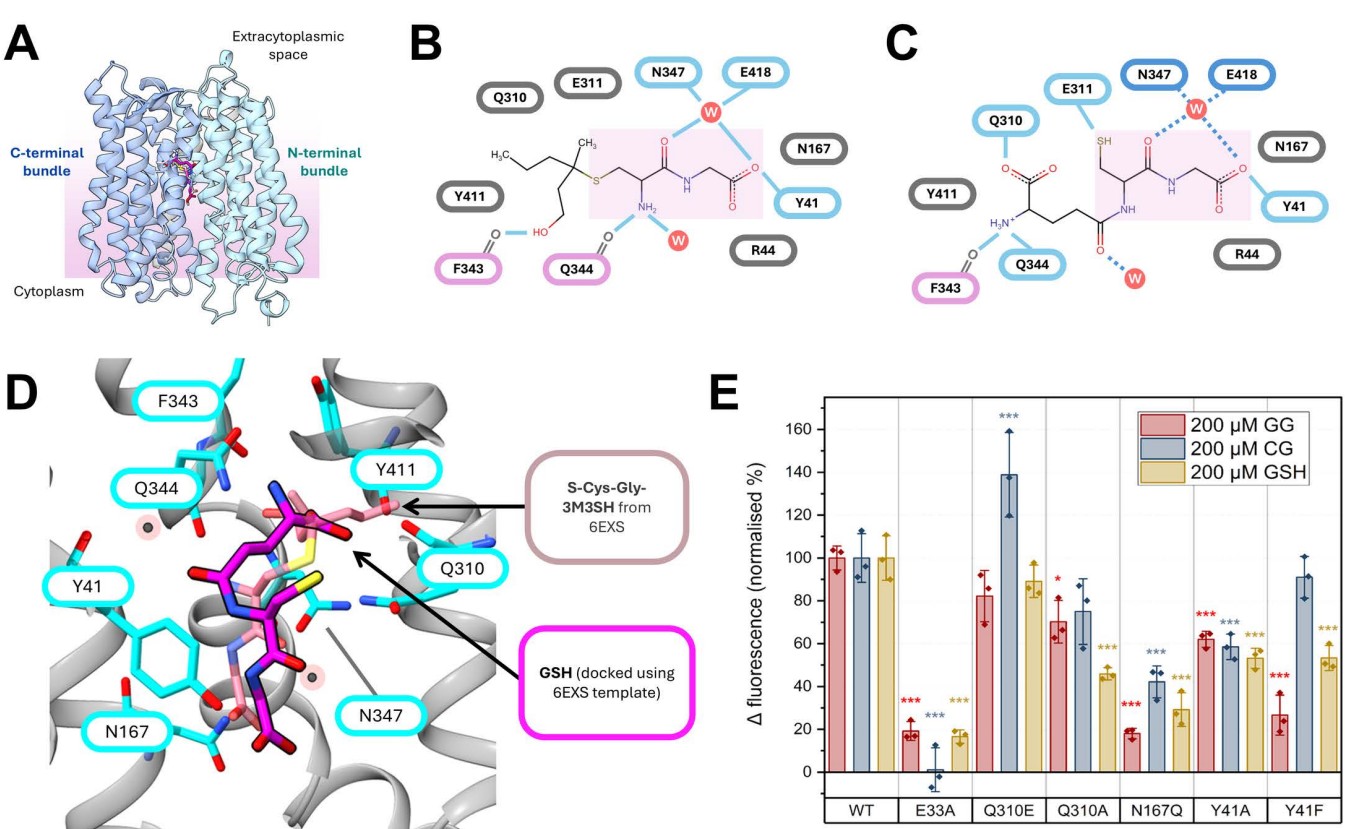

**Fig 7. Molecular docking predicts a vertical binding mode for GSH in DtpT.** (**A**) A structural model of DtpT in the inward-open conformation was generated with high confidence (QMEANDisCo Global = 0.87 ± 0.05, suggesting high per-residue quality for the model against the PepT$_{Sh}$ template). GSH (pink) was docked into the DtpT model using the existing S-Cys-Gly-3M3SH binding conformation (light pink) as a template (docking score = -5.0 kcal/mol). (**B-C**) Diagrammatic overview of protein-ligand interactions underpinning substrate coordination between PepT$_{Sh}$ and S-Cys-Gly-3M3SH (**B**) and the equivalent interactions between DtpT and GSH (**C**) in the proposed conformation. Solid blue lines indicate hydrogen bonds and polar contacts with sidechains (blue border) or main-chain groups (pink border). Dashed, dark-blue lines in (**C**) indicate possible contacts with water molecules (W). (**D**) Zoomed-in view of the proposed GSH binding mode in DtpT (grey ribbons). The known S-Cys-Gly-3M3SH binding conformation is shown for comparison. Residues surrounding the predicted binding site are shown as sticks (cyan) and labelled. (**E**) Transport data for DtpT mutant variants against GSH and two dipeptides, expressed as a percentage of the activity for the wild type (WT) protein. Bars indicate the mean ± standard deviation. *p < 0.05, ***p < 0.005; one-way ANOVA with Tukey honestly significant difference (HSD) test (against WT).

position to the equivalent group in the S-Cys-Gly-3M3SH structure, albeit shifted slightly closer to the N-terminal bundle of the protein such that the γ-glutamyl group is fully accommodated within the extended 3M3SH-binding pocket (Fig 7D). In this position, the glutamate amino group is coordinated through polar contacts with the sidechain of Gln344 and the backbone carbonyl of Phe343, while the sidechain of Gln310 sits within hydrogen bonding distance of the glutamate α-carboxyl group (Fig 7C). Glu311 also sits adjacent to the substrate thiol group and may be involved in stabilising this group via hydrogen bonding or electrostatic interactions. For both substrates, the glycyl carboxyl terminus is anchored via a hydrogen bond with the sidechain of Tyr41 and sits adjacent to Asn167, both of which are conserved across diverse POT proteins (S7 Fig). It is worth noting that the docked GSH conformation shown here does not take into account the presence of water molecules in the binding cleft, which were previously implicated in coordinating the L-Cys-Gly moiety of S-Cys-Gly-3M3SH (Fig 7B) and may similarly interact with the substrate in the proposed GSH binding conformation (as depicted in Fig 7C).

In order to assess the validity of this predicted binding model, mutant variants were generated with alterations in residues predicted to be involved in GSH binding and transport (Figs 7E and S8). Glu33 forms part of the conserved E[33]xxERFxYY[41] motif found ubiquitously in POT transporters and is known to contribute toward proton coupling during the transport cycle [27,40,41]. Unsurprisingly, an E33A mutant was found to be completely inactive against GSH and two dipeptide substrates, consistent with previous results in other POT family members (Figs 7E and S8). Substitution of Tyr41 to either alanine or phenylalanine caused a significant decrease in GSH transport, consistent with the expected role of this residue in coordinating the substrate carboxy terminus. Similarly, substitution of Asn167 to glutamine strongly attenuated transport for all tested substrates, presumably by occluding the binding site (Fig 7E). GSH transport was disproportionately attenuated in a Q310A mutant variant when compared to either Gly-Gly or Cys-Gly, supporting the proposed vertical binding conformation of this tripeptide (Fig 7E). Conversely, substitution of this residue for glutamate (Q310E) did not significantly attenuate GSH transport (Fig 7E). Overall, our findings suggest that that GSH may be accommodated by DtpT in a similar conformation to S-Cys-Gly-3M3SH in the existing PepT$_{Sh}$ complex structure, and this may indicate a unique mechanism by which POTs recognise sulphur-containing tripeptides.

## Discussion

The promiscuous nature of peptide transport proteins and their functional redundancy in many organisms have made these systems historically challenging to characterise regarding substrate range and preference. Molecular dynamics simulations and similar computational methods have been combined with biochemical and structural methods in order predict substrate preferences in POTs, but experimental evidence for the applicability of these models to biological systems is lacking [16,42,43]. Here, we present an extensive cell-based analysis of peptide transporter function in *S. aureus* and apply biochemical analysis of DtpT to both verify and expand our functional understanding of this transporter. Our findings demonstrate that DtpT is highly promiscuous, evidenced by the considerable heterogeneity observed among putative transport targets identified here (summarised in S1 Table). While earlier work has focussed on the contribution of Opp3 toward the utilisation of host-derived peptides [26], our data instead highlights how DtpT serves as a major route of uptake for diverse dipeptides and tripeptides in *S. aureus*, including GSH. This broad substrate range may underpin the essentiality of DtpT previously observed in various animal infection models [23].

A number of recent studies have demonstrated how exogenous amino acids play diverse functional roles during *S. aureus* infections, serving both as key nutrient sources and environmental signals with defined regulatory consequences [44–46]. However, such studies generally do not consider the potential contribution of host-derived oligopeptides as reservoirs for these amino acids in biological environments. Our observations reveal a novel degree of substrate selectivity within the peptide transport systems on *S. aureus*, which may point toward defined roles for these systems in accumulation of specific amino acids during growth.

For instance, glutamate is known to serve as a major gluconeogenic carbon source for *S. aureus* and strains lacking the glutamate dehydrogenase GudB are attenuated under glucose-limited conditions [47]. Glutamate restriction has

pleiotropic effects on cellular physiology, ultimately resulting in increased biofilm formation *in vitro* and *in vivo* [44]. Taken together, these observations implicate exogenous glutamate as an important factor contributing to both growth and environmental sensing in *S. aureus*. Acquisition of exogenous glutamate in *S. aureus* is primarily facilitated by the amino acid permease GltS [48]. Here, we demonstrate that DtpT is responsible for the utilisation of diverse glutamate-containing dipeptides and show that these peptides are specifically enriched among DtpT substrates (Fig 3A). These findings suggest that DtpT-mediated peptide uptake may serve as an additional source of exogenous glutamate in peptide-rich growth environments.

Our PM data reinforces the fact that DtpT serves as the primary route of dipeptide uptake in *S. aureus*, but also reveals some overlap in substrate specificity between this transporter and Opp3. The identification of a subset of peptides with redundant routes of internalisation serves as evidence that these peptides are of high value to the cell. Specifically, our data demonstrates how diverse arginine-containing dipeptides are utilised by *S. aureus* via either DtpT or Opp3 (Figs 2 and 3B). *S. aureus* USA300 rapidly internalises arginine during growth but converts neither proline nor glutamate to arginine when grown *in vitro*, making exogenous arginine essential for growth [47,32]. Experimental evidence has demonstrated how arginine restriction stimulates antibiotic tolerance in *S. aureus* biofilms, implicating arginine availability as an important environmental stimulus modulating *S. aureus* cell behaviour [45]. We propose that the redundancy observed here for uptake of Arg-containing peptides likely signifies an evolutionary strategy to increase the diversity of potential arginine sources available to the cell, which agrees with the apparent importance of exogenous arginine in *S. aureus* physiology. Furthermore, the presence of two Arg-peptide transporters with varying substrate affinities in *S. aureus* may facilitate the utilisation of these peptides in growth environments with variable Arg-containing peptide availability.

In a similar vein, DtpT is capable of transporting the glutamine-containing dipeptide Ala-Gln (Fig 4C), but our PM data points toward the existence of a yet-unidentified additional utilisation system which is specific for glutamine-containing peptides (Fig 3A). The Opp4 transport system is an uncharacterised Opp-homologue which is widespread in *S. aureus* and is likely to have been generated via duplication of Opp3 [18]. Given their close evolutionary relationship, we propose that Opp4 is likely to be a peptide transporter like Opp3 and is, therefore, a strong candidate to fulfil the role of the alternative Gln-peptide transport system in *S. aureus*. While ACME Opp may also contribute toward peptide transport, this system is more closely related to Opp1/2 and is, therefore, more likely to play a role in metal ion transport. Overall, the existence of an alternate utilisation pathway for Gln-containing peptides may point towards a yet underappreciated role for exogenous glutamine in *S. aureus* physiology.

In contrast to the examples listed above, we demonstrate that tyrosine-containing dipeptides are scarcely utilised by *S. aureus* (Fig 3C). This is perhaps unsurprising, given that *S. aureus* JE2 lacks a complete catabolic pathway for tyrosine utilisation and does not rapidly consume exogenous tyrosine when grown in defined media [47]. Intriguingly, no DtpT-mediated transport was observed for the sole tyrosine-containing peptide (Val-Tyr) assessed here (Fig 4B-4C). Our findings therefore suggest that DtpT possesses a distinct biochemical preference toward peptides containing readily catabolised amino acids over those which lack the potential to serve as a nitrogen source to fuel growth. That being said, we do observe the utilisation of some dipeptides exclusively composed of the branched-chain amino acids (i.e., leucine, isoleucine and valine) and demonstrate that several such peptides are utilised via DtpT, despite the lack of a known catabolic pathway for these amino acids in JE2 [47]. For instance, we demonstrate that Leu-Leu is utilised robustly by JE2 and belongs to cluster 2, implicating both DtpT and Opp3 in the transport of this peptide (Fig 2). Our findings suggest that branched chain amino acids derived from short peptides can serve as a nitrogen source for JE2, though the catabolic pathways involved remain to be elucidated.

Recent evidence points toward an essential role for amino acids (specifically, collagen-derived proline) in skin colonisation [4,46], and it has previously been suggested that host-derived peptides serve as an important source for these amino acids during infection [26]. While we do not observe a clear preference for Pro-containing peptides among DtpT or Opp3 substrates, we do demonstrate preferable transport of Gly-Pro by DtpT in our transport assay (Fig 4C). Hence, our

findings implicate DtpT in the utilisation of a collagen degradation product, and demonstrate how DtpT may also contribute toward proline accumulation *in vivo*.

Glutathione is a ubiquitous metabolite in eukaryotic systems and is the predominant thiol present in intracellular niches, as well as being enriched in other environments associated with *S. aureus* infection including the sputum of cystic fibrosis patients [49]. In most biological contexts, GSH is the most abundant form of glutathione by a significant margin [50]. Here, we identify GSH as a substrate of DtpT and demonstrate that either DtpT or Gis-mediated transport is sufficient to meet the nutrient sulphur requirements of *S. aureus* at physiologically relevant GSH concentrations (Fig 5). Generally, binding-protein dependant transport systems (including ABC transporters) operate with high affinities, whilst POTs and similar MFS systems have been reported as having substrate affinities in the high-micromolar or low millimolar range [16,43]. We, therefore, predict that Gis may serve as the dominant system involved in GSH scavenging under conditions where the metabolite is scarce, whilst both systems likely contribute to transport in GSH-enriched environments. Indeed, the phenotypes associated with our mutant strains support such a model (Fig 5C). The ability of DtpT to transport GSH (but not GSSG) is reminiscent of a similar role recently attributed to the Opp transport system of *E. coli* [51]. In both cases, redundant routes of GSH uptake are indicative of an important role for this form of the metabolite in relevant growth environments, and this redundancy may represent a mechanism by which the cell can tightly regulate GSH accumulation in response to environmental availability.

Finally, we identify *S. aureus* glutathione transporters as important fitness determinants during intracellular survival inside macrophage cells (Fig 6), representing an environment in which GSH is present at millimolar concentrations [38]. In *Mycobacterium tuberculosis* (Mtb), glutathione transport has been implicated in regulating the activity of Mtb-infected macrophages and thereby contributing toward intracellular survival and immune evasion [39]. Our macrophage infection data similarly demonstrates how the presence of glutathione transporters affects both bacterial survival (Fig 6) and host-cell behaviour, seemingly allowing the pathogen to modulate innate immune signalling responses and subvert premature host-cell death (S5 Fig).

To the best of our knowledge, our findings represent the first evidence of POT-mediated GSH transport in a bacterium. Whether or not this function is conserved in the wider POT superfamily remains unclear, especially given that Gln310 (or Glu in the equivalent position) is seemingly required for strong transport of GSH (Fig 7E), but this feature is not well conserved across alternative POT proteins studied to-date (S7 Fig). One exciting possibility is that POT-mediated GSH transport is a conserved trait utilised by Gram-positive bacteria to assimilate nutrient sulphur in intracellular environments. *L. monocytogenes* is perhaps the most well characterised Gram-positive intracellular bacterial pathogen, and glutathione acquisition is a known determinant of fitness in this bacterium [52,53]. Similarly to *S. aureus*, *L. monocytogenes* possesses an ABC glutathione import system (Ctp), but mutants of this system maintain the ability to utilise exogenous GSH at millimolar concentrations [52]. *L. monocytogenes* also possesses a poorly characterised POT protein (LMON_0555 in *L. monocytogenes* EGD; Fig 8A) with approximately 50% identity to DtpT, and this protein has previously been associated with fitness in a murine liver infection model [54]. Notably, all residues predicted to contribute toward GSH binding in DtpT are conserved in LMON_0555, with the exception of Gln310 which is substituted for a Glu residue in the equivalent position (Fig 8B). In DtpT, this substitution did not significantly affect GSH transport (Fig 7E). Hence, we speculate that LMON_0555 is likely to serve as a secondary glutathione acquisition system in *L. monocytogenes*, and this activity may be relevant during bacterial survival *in vivo*. More broadly, GSH transport may represent a conserved strategy utilised by Gram-positive bacterial pathogens in order to enable intracellular survival, thereby avoiding exposure to extracellular stresses and immune clearance. More work is now needed to better understand how host-pathogen interactions are affected as a result of bacterial glutathione transport and determine whether these responses vary in different bacteria and host-cell types.

Overall, this work has significantly expanded our knowledge of peptide utilisation in *S. aureus* and shed light on a novel role for glutathione transport in host-pathogen interactions during intracellular infection. However, with the diversity of

PLOS Pathogens

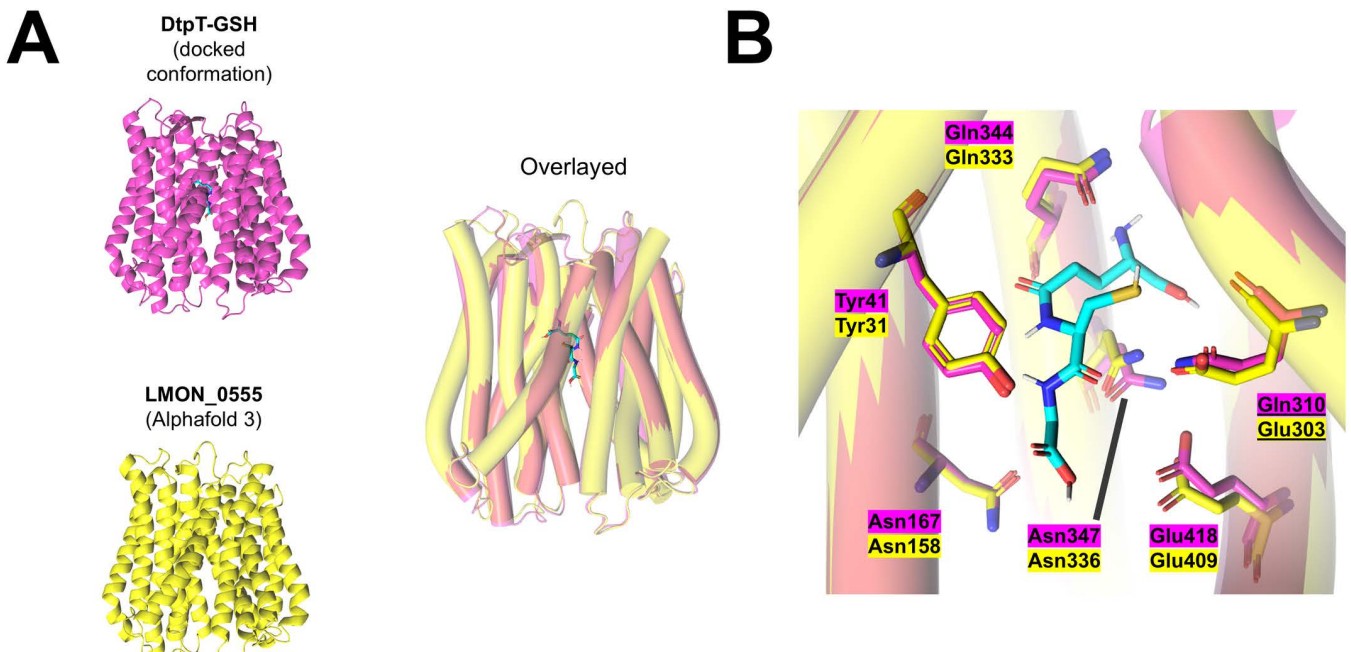

**Fig 8. AlphaFold 3 predicts conservation of the proposed GSH binding site between DtpT and LMON_0555.** (**A**) (left) Comparison of the previously generated structural model of DtpT (pink) with an AlphaFold 3 predicted structure of LMON_0555 (Yellow). The predicted binding conformation of GSH (cyan) in DtpT is also shown. (right) Overlaying these structures reveals a high degree of structural conservation (helices are depicted as cylinders to facilitate structural comparison). (**B**) Residues forming the proposed GSH binding site in DtpT and equivalent residues in LMON_0555 are shown as sticks and labelled in each case.

roles ascribed to peptide transporters in other organisms – including cell-wall turnover [11,55,56], antimicrobial peptide resistance [57,58] and cell signalling [59–61] – it is clear that this is an area of *S. aureus* biology where much remains to be discovered and understood.

## Methods

### Ethics statement

Animal experiments were conducted in accordance with the recommendations in the Guide for the Care and Use of Laboratory Animals of the National Institutes of Health. The approved protocol, PROTO202200474, was reviewed by the Animal Care and Use Committee at Michigan State University.

The Biology Ethics Committee of the University of York approved the use of National Health Service (NHS) blood cones (to DB, Ethic approval DB202111). Blood was obtained from the NHS blood service from anonymous healthy donors.

### Bacterial strains and growth conditions

All *S. aureus* strains used in this study were routinely cultured in tryptic soy broth (TSB) with shaking or on solid tryptic soy agar (TSA) medium at 37°C. *E. coli* strains were routinely cultured in lysogeny broth (LB) with shaking or on LB agar at 37°C. Selective antibiotics were added during growth when appropriate at the following concentrations: 50 μg/ml kanamycin, 100 μg/ml ampicillin, 10 μg/ml erythromycin, 10 μg/ml chloramphenicol.

*S. aureus* JE2 transposon mutants were acquired from the Nebraska Transposon Mutant Library (NTML) [62] and transduced into wild-type JE2 via Φ85 bacteriophage in order minimise off-target genetic inconsistencies between strains.

To generate double mutant strains, the Erm$^R$ resistance cassette was exchanged with an unmarked sequence using the allelic exchange vector pTnT, as described previously [63]. Plasmids were generated using the ClonExpress II One Step Cloning Kit (Vazyme) according to manufacturer's instructions. DtpT complementation in *S. aureus* mutant strains was achieved via introduction of the entire *dtpT* locus into the *E. coli/ S. aureus* shuttle vector pSK5630, modified by the addition of the native *dtpT* promoter sequence (herein referred to as pSK56$_{pT}$) [64].

A full list of bacterial strains utilised in this work are provided in S2 Table. Primers utilised for diagnostic PCR, cloning and sequencing of genetic elements are provided in S3 Table.

## Phenotype microarrays and analysis

Peptide utilisation by *S. aureus* strain JE2 and mutant derivatives was assessed on phenotype microarray plates PM 6–8 (Biolog) using a modified version of the manufacturer's protocol. Cells grown on TSA overnight were transferred from solid media into 3 ml IF-0α buffer using a sterile cotton swab and pelleted. Cells were washed twice and resuspended in IF-0α. This cell suspension was then diluted to a final OD$_{600}$ of 0.004 in 9.975 ml IF-0α and supplemented with 110 µl redox dye H (Biolog) and 915 µl 12X PM additive solution (240 mM tricarballylic acid, pH 7.1; 24 mM MgCl$_2$; 12 mM CaCl$_2$; 150 µM L-cystine, pH 8.5; 0.06% w/v yeast extract; 0.06% v/v tween 80; 6 mM D-glucose; 12 mM pyruvate). PM plates were inoculated with 100 µl/well of the prepared cell suspension and incubated for 36 hours at 37°C with continuous slow shaking in an Epoch2 microplate spectrophotometer (Agilent). OD$_{590}$ and OD$_{750}$ were measured every 15 minutes.

Empirical area under the curve (eAUC) was calculated using Growthcurver [65]. Signal curves (OD$_{590}$ - OD$_{750}$ against time) for all four strains were grouped for each peptide and then analysed by Uniform Manifold Approximation and Projection (UMAP) using the UMAP R library and a random seed of 888 [66]. K-means clustering was used on the same grouped data with "kmeans" R function and k = 7. The value of k = 7 was selected based on preliminary analysis of the underlying data, in which it was noted that seven distinct utilisation patterns were represented across the majority of the tested peptides.

## Peptide utilisation assays

Utilisation of specific peptides was assessed using glucose-supplemented chemically defined medium (CDMG) based on a previously reported methodology [67]. The complete composition of the CDMG used here is provided in S4 Table. For assessing the utilisation of arginine-containing peptides, L-arginine was omitted from the medium (CDMG - *R*). For assessing the utilisation of glutathione, L-cysteine and L-methionine were omitted from the medium and (NH$_4$)$_2$SO$_4$·FeSO$_4$·6H$_2$O was substituted for 15 µM iron (III) citrate (CDMG - *s*). Reduced glutathione solutions were prepared under anaerobic conditions to prevent changes in the oxidation state before commencing assays. Overnight cultures of *S. aureus* strain JE2 and mutant derivatives were diluted 1:100 into fresh TSB and grown to mid-exponential phase. Cells were then pelleted and washed twice in the assay medium (CDMG - *R* or CDMG - *s*) before being resuspended and diluted to a final OD$_{600}$ of 0.05 in peptide-supplemented medium. 200 µl of each cell suspension was transferred to a Costar 96 well plate (Corning) and incubated for 24 hours at 37°C with shaking. OD$_{600}$ was measured every 30 minutes.

For anaerobic growth assays, cultures were prepared as before being transferred to a Whitley A85 anaerobic workstation (Don Whitley Scientific) and diluted into assay medium which had been equilibrated under anaerobic conditions for at least 3 hours and supplemented with 100 mM sodium nitrate. Growth was monitored as above in a Stratus 96 well plate reader (Cerillo) for 18 hours at 37°C without shaking.

## Protein purification

The DtpT open reading frame was amplified directly from the *S. aureus* JE2 genome and cloned into the C-terminal octa-histidine/GFP-tagged fusion vector pWaldo [68] before being transformed into *E. coli* C43(DE3) [69]. Mutant variants

were generated via site-directed PCR mutagenesis and blunt-end ligation with T4 DNA ligase (NEB) and all constructs were verified by sequencing.

Cells grown overnight in LB supplemented with kanamycin were diluted 1:100 into fresh media. Cultures were incubated at 37°C until an $OD_{600}$ of 0.5 was reached. Expression was then induced by the addition of IPTG to a final concentration of 100 µM. Cultures were incubated overnight at 25°C with shaking. Cells were then pelleted, resuspended in 50 ml resuspension buffer (40 mM Tris, pH 7.5; 10% glycerol) and lysed by sonication. Crude lysates were clarified via centrifugation at 27,000 x g for 30 minutes. Membranes were then pelleted via ultracentrifugation at 164000 x g for 120 minutes. Membrane pellets were pooled and resuspended in 40 ml protein solubilisation buffer (40 mM Tris, pH 7.5; 200 mM NaCl; 20 mM imidazole; 10% glycerol; 0.5% n-dodecyl-B-pyromaltoside (DDM)) using a glass homogenizer before being incubated for 60 minutes at 4°C with rolling. The resultant suspension was then ultracentrifuged as before for 60 minutes to pellet leftover membranes and insoluble contaminants. Supernatant containing the solubilised membrane proteins was pooled and applied to a His-TRAP nickel affinity column (GE Healthcare) using an ÄKTA Protein Purification System (Cytiva). This column was washed with 50 ml column wash buffer (40 mM Tris, pH 7.5; 200 mM NaCl; 20 mM imidazole; 10% glycerol; 0.04% DDM) and the fusion protein was then eluted via application of column elution buffer (40 mM Tris, pH 7.5; 200 mM NaCl; 400 mM imidazole; 10% glycerol; 0.04% DDM).

Purified protein was exchanged into TEV reaction buffer (40 mM Tris, pH 7.5; 200 mM NaCl; 0.5 mM EDTA; 1 mM DTT; 10% glycerol; 0.04% DDM) using a HiTRAP desalting column (SLS). Protein was then mixed with TEV protease to a final ratio of approximately 3:1. The resultant reaction mixture was incubated overnight at 4°C. Pure DtpT was isolated by applying the reaction mixture to a His-TRAP nickel affinity column as before and washing with column wash buffer, this time collecting the flow-through fractions. Pure protein was stored at -70°C.

## Reconstitution into proteoliposomes and transport assays

Purified DtpT was exchanged into reconstitution buffer (20 mM Tris pH 7.5; 150 mM NaCl; 0.03% DDM) and diluted to a final concentration of 0.5 mg/ml. Liposomes (composed of POPE and POPG in a 3:1 ratio) were prepared and extruded 11 times each through pre-soaked 0.8 µm and 0.4 µm track-etched polycarbonate membranes (Whatman) before diluting to a final concentration of 10 mg/ml in liposome buffer (50 mM KPi, pH 7).

10 mg lipid mixture was gradually added to 200 µg protein at room temperature before being incubated on ice for 60 minutes. 100 µl SM2 Biobeads (Biorad) suspended in water were added to the lipid-protein mixture, which was then incubated for 1 hour at 4°C while rotating. A further 130 µl Biobeads were added and the mixture was incubated as before for a further 3 hours. Biobeads were pelleted and the lipid-protein mixture was transferred to a fresh tube. 160 µl Biobeads were added and the mixture was incubated as before for a further 16 hours. Biobeads were removed as before and proteoliposomes were pelleted via ultracentrifugation at 158 000 × g for 30 minutes. Proteoliposomes were then resuspended in liposome buffer to a final protein concentration of 0.5 mg/ml (assuming 100% reconstitution efficiency) and subjected to three freeze-thaw cycles before dialysing extensively against liposome buffer at 4°C. Proteoliposomes were pelleted as before and resuspended in liposome buffer before being stored at -70°C.

For transport assays, proteoliposomes were pelleted as before and resuspended in inside buffer (5 mM HEPES, pH 6.8; 120 mM KCl; 2 mM $MgSO_4$) spiked with 1 mM pyranine. Proteoliposomes were subjected to seven freeze-thaw cycles before being extruded through a pre-soaked 0.4 µm track-etched polycarbonate membrane. Proteoliposomes were then pelleted at 20°C and resuspended in inside buffer. Excess pyranine was removed using a microspin G25 desalting column (Cytiva) according to the manufacturer's protocol. Liposomes were finally pelleted again at 20°C and resuspended to a final protein concentration of 1 mg/ml. For each assay, 6 µg of proteoliposomes were diluted 1:100 into outside buffer (5 mM HEPES, pH 6.8; 120 mM NaCl; 2 mM $MgSO_4$). Assays were performed in 1 ml clear plastic microcuvettes using a Fluoromax-4 spectrophotometer (Horiba) with a magnetic flee for stirring. Fluorescence of pyranine was measured via excitation at 460 nm and emission at 510 nm. Substrate was added to the reaction at t = 25 s and transport was induced by

the addition of 1 μM valinomycin at t = 50 s. All curves are normalised before the addition of valinomycin (t = 45 s) to facilitate comparison.

## Macrophage culture and differentiation

THP1 cells (ATCC) were cultured in macrophages media (RPMI1640 supplemented with 10% Fetal Calf Serum and Glutamine). To obtain a macrophage-like differentiation state, THP1 were differentiated for 48 hrs in macrophage media supplemented with 50 ng/mL Phorbol 12-myristate 13-acetate (in DMSO, Sigma-Aldrich). PMA-containing media was replaced with fresh macrophage media and cells were rested overnight prior to infection.

Human primary macrophages were differentiated from human peripheral blood mononuclear cells (PBMCs) obtained from the NHS (University of York Biology ethic committee project DB202111) and differentiated in complete macrophage media supplemented with 50 ng/mL macrophage colony-stimulating factor (mCSF, Proteintech) over 7 days.

## Macrophage infection

Infection of macrophages and macrophage-like cell lines was carried out based on previously published methods [70]. Overnight cultures of *S. aureus* strain JE2 and mutant derivatives were diluted 1:100 into fresh TSB and grown to mid-exponential phase. Cells were then pelleted and washed twice in ice-cold phosphate-buffered saline (PBS) before being resuspended to a final cell density of $5 \times 10^7$ CFU/ml. For infection, 10 μl bacterial suspension was added to 50,000 hMDM or THP-1 cells (D.O.I. = 10) in 90 μl Opti-MEM reduced serum media (Thermo Fisher) and plates were centrifuged for 5 minutes at 500 x g to promote contact. After 30 minutes of incubation at 37°C, the medium was exchanged for Opti-MEM supplemented with 100 μg/ml gentamicin. The addition of gentamicin was designated t = 0 h. After a further 30 minutes of incubation this medium was removed and cells were then maintained in Opti-MEM supplemented with 5 μg/ml gentamicin (Gibc).

To compare bacterial internalisation and survival, cells at t = 0.5/ 8 h were incubated in deionised water for 10 minutes before being lysed by repeated pipetting. Suspensions were serially diluted in PBS, spotted onto TSA and incubated overnight at 37°C to determine viable CFU. Cytotoxicity was measured by quantifying the activity of lactate dehydrogenase (LDH) in the extracellular medium (Cytotox 96 non-radioactive cytotoxicity kit, Promega) and normalised against cells lysed by 0.04% Triton X-100. Cytokine levels in cell culture medium were determined at t = 6 h by utilising human IL 1-β, IL 6 and TNF-α ELISA assay kits (invitrogen) according to the manufacturer's protocols.

## Structural modelling and docking

A structural model of DtpT was generated using SWISS-MODEL [71] using the solved complex structure of the *S. hominis* PepT$_{Sh}$ (6EXS) as a template. For predicting how DtpT recognises glutathione, blind docking was carried out using this projected structure in CB-Dock 2 [72]. A high confidence predicted structure of LMON_0555 (pTM = 0.94) was generated using Alphafold 3 [73]. Projected and docked structures were visualised in ChimeraX (UCSF).

## Supporting information

**S1 Fig. Arginine-containing dipeptides restore growth under arginine limitation only when a viable route of uptake is available.** (A) Growth of *S. aureus* strain JE2 and mutant derivatives in CDMG, CDMG – R and CDMG – R supplemented with three dipeptides, as labelled. Corresponding PM clusters are also provided. (B) Growth of deficient strains is restored by *in-trans* expression of *dtpT* under its native promoter in each case. In each case, strains were grown over 24 hours and $OD_{600}$ was measured every 30 minutes for three biological replicates. Curves indicate the mean values for each reading and the standard deviation in each case is indicated by the shaded area.
(TIF)

**S2 Fig. Purification of DtpT from *E. coli* C43 (DE3).** (**A**) Nickel affinity purification of a DtpT-GFP fusion protein. Images correspond to total protein (**left**; Coomassie blue stain) and DtpT-GFP only (**right**; in-gel GFP fluorescence). Mem = Membrane fraction. F.T = Column flow-through. Conc = concentrated fractions. (**B**) Representative gel image of a reverse nickel affinity purification of untagged DtpT after TEV cleavage of GFP-8His. (**C**) SEC purification of pure DtpT protein following TEV cleavage. A single sharp peak is seen at approximately 12.5 ml elution volume in the A280 trace (**upper**) corresponding to the pure DtpT protein, split across elution fractions 24–26 (**lower**). Overall, approximately 2.35 mg of pure DtpT protein was yielded from 2 L of bacterial culture as estimated from A280. The black arrow in each case indicates the position of a band corresponding to the pure DtpT protein. The red arrow indicates the position of a band corresponding to the expected DtpT-GFP fusion protein. The striped arrow indicates the position of sfGFP.
(TIF)

**S3 Fig. DtpT proteoliposomes require both a valid substrate and potential gradient in order to drive proton-coupled transport.** (A) Successful reconstitution of DtpT was confirmed by size exclusion chromatography (left) and SDS PAGE (right). The expected position of the soluble DtpT protein is indicated by the black arrow. (B) CD spectra of pure DtpT (grey) and DtpT liposomes (pink). A scaling factor of 2x has been applied to the DtpT-liposome spectrum to facilitate comparison with the detergent-soluble protein. (C) Validation of pyranine assays in DtpT liposomes. Curves show the mean of three replicates normalised to the initial fluorescence signal (t = -50 s). In each case, peptide (or $H_2O$) is added at t = -25 s (blue arrow) and a negative-inside $\Delta\Psi$ is established by the addition of valinomycin at t = 0 s (red arrow). No acidification is observed in the absence of DtpT. A slight decrease in fluorescence is observed upon addition of valinomycin in the absence of substrate, likely due to proton leakage. Strong acidification of the lumen indicative of DtpT-mediated transport requires both peptide substrate and $\Delta\Psi$.
(TIF)

**S4 Fig. Extended growth assays provide further insight into the routes of glutathione transport in *S. aureus*.** (**A**) Growth of *S. aureus* strain JE2 carrying pSK56$_{pT}$ (Empty) and mutant derivatives carrying either pSK56$_{pT}$ or pSK56$_{pT}$*dtpT* (Comp) was assessed in CDMG – *s* supplemented with 1 mM GSH and chloramphenicol. (**B** - **C**) Growth of *S. aureus* strain JE2 and mutant derivatives was assessed in CDMG – *s* supplemented with 250 μM GSSG (**B**) in aerobic conditions, as well as in the presence of 1 mM GSH under anaerobic conditions (**C**). For A and B, strains were grown over 24 hours and $OD_{600}$ was measured every 30 minutes. For C, strains were grown over 18 hours and $OD_{600}$ was measured every 15 minutes. Curves indicate the mean values for three biological replicates ± standard deviation. For B and C, a pale grey line indicates equivalent data for strain JE2 in CDMG – *s* and is included for comparison.
(TIF)

**S5 Fig. Glutathione transporters modulate the host response to infection inside THP-1 cells.** (**A** - **B**) Cell death following bacterial infection was quantified at 4 hours (**A**) and 8 hours (**B**) post-infection for THP1 cells infected with JE2 or mutant derivatives (n = 6, each measured in technical triplicate). Cell death was quantified by measuring the activity of extracellular LDH in cell culture supernatant. Values are given as a percentage of the LDH activity for cells lysed by addition of 0.04% triton X-100 and corrected for background signal. (**C**) Production of inflammatory cytokines by infected THP-1 cells at 6 hours post-infection (n = 4, each measured in technical triplicate). *p < 0.05, **p < 0.01; as determined by paired-sample t-test (vs JE2).
(TIF)

**S6 Fig. Virulence quantification of *dtpT*::Tn and Δ*gis dtpT*::Tn mutants in a murine model of systemic infection.** (A - B) Bacterial burdens within indicated organs after systemic inoculation of BALB/cJ mice were enumerated after 96 h of infection with either WT (squares), *dtpT*::Tn (circles) (**A**) or Δ*gis dtpT*::Tn (diamonds) (**B**). Bacterial burdens are presented as $\log_{10}$ transformed CFUs mL$^{-1}$ for liver, combined kidneys, and heart. The mean and standard deviation are presented as horizontal

lines. Normality was determined using a Shapiro-Wilk test. *p* values were determined by Mann-Whitney test. (**C**) Competitive indices (CI) for *dtpT*::Tn, *gisB*::Tn, and Δ*gis dtpT*::Tn were determined from the livers of Balb/cJ mice systemically inoculated with a 1:1 mixture of WT and the indicated mutant strain. The liver output ratio of WT to indicated mutant CFU mL$^{-1}$ in the liver over the input WT to mutant CFU mL$^{-1}$ were used to calculate the CI. The mean of WT:indicated mutant CI are presented as a horizontal bar and *p* values denoted for each competition were determined by Wilcoxon signed-ranked test. (TIF)

**S7 Fig. Multiple sequence alignment of DtpT and structurally characterised POTs.** Multiple protein sequence alignment was carried out by Clustal Omega [Madeira et al., 2024] and visualised in Jalview [Waterhouse et al., 2009]. The sequence of DtpT (from *S. aureus* JE2) is highlighted. Sequences are given in FASTA format and coloured by identity (blue). Residues predicted to contribute toward GSH binding in DtpT are highlighted by red boxes. Regions corresponding to predicted structural elements in DtpT are also labelled above each line. (TIF)

**S8 Fig. Complete transport assay data for DtpT and mutant variants compared to a water-only control.** (**A**) SDS PAGE comparison of pure DtpT against solubilised liposomes containing wild-type or mutant variants of DtpT, as labelled. Band intensities were used to normalise the final liposomal protein concentration to 0.5 mg/ml. (**B**) Complete transport assay data for each of three peptide substrates, normalised immediately before the addition of valinomycin for ease of comparison. Curves show the mean of three replicates. (**C**) Summarised transport data for DtpT and mutant variants against gly-gly, cys-gly and GSH. Transport activity is compared by quantifying the change in pyranine fluorescence over the first 30 seconds of the assay. A grey dashed line indicates the Δ fluorescence (%) recorded for DtpT liposomes in the absence of substrate (water only). Bars indicate the mean ± standard deviation. (TIF)

**S1 Table. Summary of DtpT substrates identified in this work.** (PDF)

**S2 Table. Strains utilised in this work.** (DOCX)

**S3 Table. Primers utilised in this work.** (DOCX)

**S4 Table. Composition of CDMG (adapted from [67]).** (DOCX)

**S1 File. Methods.** Supplemental methods supporting S6 Fig. (DOCX)

**S1 Data. Supplementary data file containing raw data supporting** Figs 1, 3, 4, 5, **S6 and S7.** (XLSX)

**S1 Dataset. PM 6 full data w legend.** (JPG)

**S2 Dataset. PM 7 full data w legend.** (JPG)

**S3 Dataset. PM 8 full data w legend.** (JPG)

**S4 Dataset.  SD4 eAUC summary for PM clusters.**
(XLSX)

## Acknowledgments

The defined transposon mutant library used in this study was provided by the Network on Antimicrobial Resistance in Staphylococcus aureus (NARSA) for distribution by BEI Resources, NIAID, NIH: Nebraska Transposon Mutant Library (NTML) Screening Array NR-48501. We thank Dr Rebecca M. Corrigan (ORCiD: 0000-0002-6031-1148) for supplying the wild-type JE2 and NTML strains utilised in this study, and for providing valuable technical guidance. We also thank Dr Christopher Mulligan (ORCiD: 0000-0001-5157-4651) and his lab for their technical guidance in regard to membrane transporter protein techniques utilised in this work.

## Author contributions

**Conceptualization:** Imran Khan, Neal D Hammer, Marjan van der Woude, Gavin H. Thomas.

**Data curation:** Imran Khan.

**Formal analysis:** Imran Khan, Sandy J MacDonald, Simon Newstead, Dave Boucher, Marjan van der Woude, Gavin H. Thomas.

**Funding acquisition:** Marjan van der Woude, Gavin H. Thomas.

**Investigation:** Imran Khan, Sigurbjörn Markússon, Paige J Kies, Cristina Kraemer-Zimpel, Callum Robson, Joanne L Parker, Simon Newstead, Dave Boucher.

**Methodology:** Imran Khan, Marjan van der Woude.

**Project administration:** Marjan van der Woude, Gavin H. Thomas.

**Software:** Sandy J MacDonald.

**Supervision:** Joanne L Parker, Simon Newstead, Dave Boucher, Neal D Hammer, Marjan van der Woude, Gavin H. Thomas.

**Writing – original draft:** Imran Khan, Joanne L Parker, Marjan van der Woude, Gavin H. Thomas.

**Writing – review & editing:** Joanne L Parker, Simon Newstead, Dave Boucher, Neal D Hammer, Marjan van der Woude, Gavin H. Thomas.

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
