## [Decision Letter · Decision Letter 0]

24 Jun 2025

Defined roles for the Staphylococcus aureus POT transporter DtpT in di/tripeptide uptake and glutathione utilisation inside human macrophages

PLOS Pathogens

Dear Dr. Thomas,

Thank you for submitting your manuscript to PLOS Pathogens. After careful consideration, we feel that it has merit but does not fully meet PLOS Pathogens's publication criteria as it currently stands. Therefore, we invite you to submit a revised version of the manuscript that addresses the points raised during the review process.

Please submit your revised manuscript within 60 days Aug 23 2025 11:59PM. If you will need more time than this to complete your revisions, please reply to this message or contact the journal office at plospathogens@plos.org. Please include the following items when submitting your revised manuscript:

We look forward to receiving your revised manuscript.

Kind regards,

Andreas Peschel, Ph.D.

Academic Editor

PLOS Pathogens

Anne Jamet

Section Editor

PLOS Pathogens

Editor-in-Chief

PLOS Pathogens

orcid.org/0000-0003-2946-9497

Michael Malim

PLOS Pathogens

orcid.org/0000-0002-7699-2064

**Journal Requirements:**

2) Please insert an Ethics Statement at the beginning of your Methods section, under a subheading 'Ethics Statement'. It must include:

i) The full name(s) of the Institutional Review Board(s) or Ethics Committee(s)

ii) The approval number(s), or a statement that approval was granted by the named board(s)

iii) A statement that formal consent was obtained (must state whether verbal/written) OR the reason consent was not obtained (e.g. anonymity). NOTE: If child participants, the statement must declare that formal consent was obtained from the parent/guardian.].

5) We note that your Data Availability Statement is currently as follows: "There is no omics and structural data in this paper and all primary data is provided in figures and Tables.". Please confirm at this time whether or not your submission contains all raw data required to replicate the results of your study. Authors must share the “minimal data set” for their submission. PLOS defines the minimal data set to consist of the data required to replicate all study findings reported in the article, as well as related metadata and methods (https://journals.plos.org/plosone/s/data-availability#loc-minimal-data-set-definition).

**Reviewers' Comments:**

Reviewer's Responses to Questions

**Part I - Summary**

Reviewer #1: These studies document that DtpT functions as a dipeptide transporter in S. aureus and transports a significant number of dipeptides including glutathione (GSH only) at higher concentrations (~1 mM). However, other dipeptides and larger oligopeptides are transporters via other specific transporters (Opp3 and unknown dipeptide transporters). These data suggest, which is quite impactful, that S. aureus has evolved to interact with specific dipeptides...and those dipeptides are transported via specific transporters. Strengths of this manuscript include the rigorous documentation of the function of DtpT with respect to the diversity of dipeptides that it can transport. This is differentiated from Opp3 that seems to transport longer peptides (3-8 mer). In addition, DtpT was shown to transport GSH but not GSSH.

Reviewer #2: The authors of the manuscript characterize the POT tansporter DtpT in the major human pathogen Staphylococcus aureus. They use a noteworthy diversity of approaches ranging from structural and biochemical methods to the role of peptide transporters in macrophage and mouse infection models. The findings further clarify the utilization by S. aureus of peptides and host supplied nutrients such as glutathione in an infection setting.

The importance of the role of metabolism and nutrient uptake under infectious and colonizing conditions of human pathogens such as S. aureus has been increasingly noted in the last years. Especially the switch from colonization to infectious lifestyles could be connected to different nutritional cues and availability in different ecological niches. Hence, I think this contribution is very welcomed and apt addition to the scientific communities. Nonetheless I am of the impression that some parts of manuscript lack overall clarity and overemphasize the significance of certain results.

**Part II – Major Issues: Key Experiments Required for Acceptance**

Reviewer #1: One of the major aspects of the manuscript is the documentation that DtpT transports glutathione but is not the major glutathione transporter (GisD). In these cases where the transporters are known, transport assays should be performed to understand the transport kinetics to more fully understand the hypothesis that GisD is a high affinity transporters whereas DtpT is a low affinity transporter (similar to Figure 7E). This would add significant rigor to the manuscript.

Unclear if the last part of the study regarding Listeria should be added to the manuscript. Maybe this should be discussed in the discussion section?

Although the assay seems to have worked, the Biolog assays are difficult to interpret at best as one does not know what types of compounds are in the wells (proprietary information--they were designed for prototrophs). For instance, as noted by the authors, S. aureus requires a significant number of amino acids for appropriate growth so nitrogen based assays are difficult to design. I am unsure what the authors can say about this as they confirmed many of the results, but, again, difficult to interpret as you can't design how the assay is run.

Reviewer #2: The authors used a commercially available phenotype microarrays (“Biolog Plates”). S. aureus however exhibits distinct metabolic peculiarities such as auxotrophies, genetic and regulatory ones, that set it apart from the model organism E. coli. These auxotrophies make conclusions of nutrient or nitrogen metabolization difficult since S. aureus metabolism has many unresolved interdependencies as certain amino acids and cofactors have to be included into minimal growth media (see Audretsch et al 2021, https://doi.org/10.1038/s41598-021-88646-1, and Machado et al 2019, https://doi.org/10.1128/AEM.01773-19). These essential growth factors often contain nitrogen themselves and could potentially be utilized as direct nitrogen source? Only cysteine was apparently added. What about other essential aminoacids? Hence several essential growth factors (amino acids or cofactors such as thiamine) can bottleneck quantifiable growth and not the supplied peptides themselves. Data in the manuscript that addresses the applicability or usability of commercial phenotype microarrays for S. aureus to extract high quality scientific results does not appear in the manuscript.

Furthermore omission of certain aminoacids which essential due to repression by Ccpa or CodY often lead to the creation of suppressor mutants. Those suppressor mutant however often have brought effects potentially on other systems such as dtpT or opp3A. Appearance of suppressor mutants usually can be detected and excluded based on their stochastic appearance. These points bring me to the following questions which I feel the authors should address. In general I think the quality of scientific information of the screening approach is strongly overemphasized.

I don’t find any biological or technical replicates mentioned by the authors used Biolog plates screen. How many were used for the screen? Were potential suppressor mutants overserved and excluded?

From the description in the method section the Biolog plates contained the amino acid cystein which can be essential in S. aureus. However several other amino acids are also essential under certain regulatory regimens. Was this tested in the setup of the screen? Also 3 different carbon sources were used in the growth medium used for the screening: pyruvate, glucose, and citric acid. Previous work showed strong effect of supplemented carbon sources on the ability to synthesize amino acids (Lit et al 2010, doi:10.1128/JB.00237-10). Have the authors considered these effects?

The authors used an unsupervised clustering algorithm for the obtained growth curves. Usually this kind of analysis should be performed if the content of the dataset is of high complexity and difficult to interpret with classical methods. I think this is not the case as the datasets seems to consist of clear growth or weak/no growth events. The authors should explain why they used K-means clustering with the K parameter set to k=7. In the end the algorithm just “presses” the data into 7 arbitrary bins. Why not k=3 or k=30? The authors make it seems that there are 7 observable distinct phenotypic outcomes which is an artifact of their data analysis pipeline. The K-means clustering already quite strongly hints that each growth curve can be interpretated directly and more simply by the respective growth values....

I would suggest approaching the screening data more actually based on the observed phenotypes from the growth curves. Each growth curve would here be classified as “no/ growth” or growth/strong growth based on the actual control datapoints (DMSO and glutamine) included in the Biolog plates. (The controls are also not shown/addressed by the authors). Hence each supplied peptide substrate can than be binned into a certain group based on the phenotypic “growth fingerprint”.

Data presentation is often not clear or consistent. Often figure legends are lacking key information. I suggest to improve the way the data is presented by using clear consistent rules for presenting the data. Figure 1 and 3 show barplots with the same set of strains but are colours not used consistently in both figures. Figure 2 uses a lot of colours to highlight the 7 k clusters. However the used palette does not appear very suitable for people with impaired colour perception. The authors should use pallets or contrasts that are suitable for people with colour vision deficiency.

I further recommend summarizing the information of the different assays performed in regards to the substrate specificity of DtpT in a table. This could be a good resource for further studies on the system.

The statistical tests applied appear arbitrary: unpaired t-test versus ANOVA with different post-tests. Why is this not consistent throughout the figures?

**Part III – Minor Issues: Editorial and Data Presentation Modifications**

Reviewer #1: What might be the biology behind the data that DtpT transports GSH but not GSSH? Do other glutathione transporters function in this manner.

Reviewer #2: Typos and style issues-

In the author summary: “fulfill”, “tree of life”

“Are widely distributed in staphylococci.”

In the sentence “To compare DtpT-mediated transport of peptides directly, we expressed and purified S. aureus DtpT recombinantly from E. coli cells (Figure S2) and...” I recommend to spell out the species name of E. coli since it was used here first in the manuscript.

Figure 1:

1D) The authors should state what a “signal curves” exactly is. What was measured here? The use of means and “error bars” is misleading as these are not biological replicates but rather individual samples so other statistics might be more appropriate here? Again it is not clear how often the experiment was repeated.

Figure 2:

The authors use 12 + colours in their figure. I suggest to reduce the number of colours for example by using other means such as shadings or patterns. Please further clarify the columns and rows of the matrix (N-term and C-terminus)

Figure 4:

Figure 4A. This figure needs a clearer legend. I suggest to specify the icons (star, double-circle etc.). Figure B and C. The color shading of the datapoints indicate the results of an ANOVA test. However there is no reference or legend to which colour is significant. Figure B why are the standard deviations omitted from the lines? In figure 4C there seems to be lack of a positive control. Is there a known and confirmed dipeptide which shows strong uptake by DtpT which could be utilized for this purpose?

Figure 6:

What do the shown boxes and whiskers indicate? This is not indicated in the figure legend.

PLOS authors have the option to publish the peer review history of their article (what does this mean? ). If published, this will include your full peer review and any attached files.

**Do you want your identity to be public for this peer review?** For information about this choice, including consent withdrawal, please see our Privacy Policy .

Reviewer #1: No

Reviewer #2: No

**Figure resubmission:**

**Reproducibility:**



---

## [Decision Letter · Decision Letter 1]

12 Sep 2025

Dear Prof. Thomas,

We are pleased to inform you that your manuscript 'Defined roles for the Staphylococcus aureus POT transporter DtpT in di/tripeptide uptake and glutathione utilisation inside human macrophages' has been provisionally accepted for publication in PLOS Pathogens.

Best regards,

Andreas Peschel, Ph.D.

Academic Editor

PLOS Pathogens

Anne Jamet

Section Editor

PLOS Pathogens

Sumita Bhaduri-McIntosh

Editor-in-Chief

PLOS Pathogens

orcid.org/0000-0003-2946-9497

Michael Malim

Editor-in-Chief

PLOS Pathogens

orcid.org/0000-0002-7699-2064

Reviewer #1:

Reviewer #2:

Reviewer Comments (if any, and for reference):

Reviewer's Responses to Questions

**Part I - Summary**

Reviewer #1: No further comments, I appreciate the discussion by the investigative team.

Reviewer #2: The authors of the manuscript characterize the POT tansporter DtpT in the major human pathogen Staphylococcus aureus. A strong point is that the authors use a noteworthy diversity of approaches ranging from structural and biochemical methods to the role of peptide transporters in macrophage and mouse infection models. The findings further clarify the utilization by S. aureus of peptides and host supplied nutrients such as glutathione in an infection setting. However, the exact role of the described DtpT transporter in amino acid and glutathione metabolism remains elusive to a certain degree as downstream processes are not insufficiently understood. However, this is out of scope for this study. This could be addressed in future studies.

The importance of the role of metabolism and nutrient uptake under infectious and colonizing conditions of human pathogens such as S. aureus has been increasingly noted in the last years. Especially the switch from colonization to infectious lifestyles could be connected to different nutritional cues and availability in different ecological niches. Hence, I think this contribution is very welcomed and apt addition to the scientific communities.

**Part II – Major Issues: Key Experiments Required for Acceptance**

Reviewer #1: None

Reviewer #2: The authors could address most questions I raised. Especially they clarified the biochemical character of their output signal of their high-throughput assay in a satisfying matter. Also, they outlined their rationale for the chosen clustering. They emphasized that the unsupervised clustering reflected the results of an initial analysis driven by phenotypic observations.

**Part III – Minor Issues: Editorial and Data Presentation Modifications**

Reviewer #1: None

Reviewer #2: The colouring of the data bars in figure 4B is still non sensical and not specified sufficiently in the figure legend. I suggest replacing the current colours with a clear colour scheme indicating the shown significance levels.

The authors should state that the shown results of the ortholog plate screened are derived from a single biological replicate. This should occur either in the method section or the legend of figure 1.

PLOS authors have the option to publish the peer review history of their article (what does this mean? ). If published, this will include your full peer review and any attached files.

**Do you want your identity to be public for this peer review?** For information about this choice, including consent withdrawal, please see our Privacy Policy .

Reviewer #1: No

Reviewer #2: No

---

## [Editor Report · Acceptance letter]

Dear Prof. Thomas,

We are delighted to inform you that your manuscript, "Defined roles for the Staphylococcus aureus POT transporter DtpT in di/tripeptide uptake and glutathione utilisation inside human macrophages," has been formally accepted for publication in PLOS Pathogens.

Best regards,

Sumita Bhaduri-McIntosh

Editor-in-Chief

PLOS Pathogens

orcid.org/0000-0003-2946-9497

Michael Malim

Editor-in-Chief

PLOS Pathogens

orcid.org/0000-0002-7699-2064